# Evolutionarily related host and microbial pathways regulate fat desaturation in *C. elegans*

Bennett W. Fox [1], Maximilian J. Helf[1,4], Russell N. Burkhardt[1,4], Alexander B. Artyukhin[2], Brian J. Curtis[1], Diana Fajardo Palomino[1], Allen F. Schroeder[1], Amaresh Chaturbedi[3], Arnaud Tauffenberger[1], Chester J. J. Wrobel[1], Ying K. Zhang[1], Siu Sylvia Lee [3] & Frank C. Schroeder [1] ✉

Fatty acid desaturation is central to metazoan lipid metabolism and provides building blocks of membrane lipids and precursors of diverse signaling molecules. Nutritional conditions and associated microbiota regulate desaturase expression, but the underlying mechanisms have remained unclear. Here, we show that endogenous and microbiota-dependent small molecule signals promote lipid desaturation via the nuclear receptor NHR-49/PPARα in *C. elegans*. Untargeted metabolomics of a β-oxidation mutant, *acdh-11*, in which expression of the stearoyl-CoA desaturase FAT-7/SCD1 is constitutively increased, revealed accumulation of a β-cyclopropyl fatty acid, becyp#1, that potently activates *fat-7* expression via NHR-49. Biosynthesis of becyp#1 is strictly dependent on expression of cyclopropane synthase by associated bacteria, e.g., *E. coli*. Screening for structurally related endogenous metabolites revealed a β-methyl fatty acid, bemeth#1, which mimics the activity of microbiota-dependent becyp#1 but is derived from a methyltransferase, *fcmt-1*, that is conserved across Nematoda and likely originates from bacterial cyclopropane synthase via ancient horizontal gene transfer. Activation of *fat-7* expression by these structurally similar metabolites is controlled by distinct mechanisms, as microbiota-dependent becyp#1 is metabolized by a dedicated β-oxidation pathway, while the endogenous bemeth#1 is metabolized via α-oxidation. Collectively, we demonstrate that evolutionarily related biosynthetic pathways in metazoan host and associated microbiota converge on NHR-49/PPARα to regulate fat desaturation.

Metazoan metabolism is intricately linked to that of associated microbiota, and host-microbial co-evolution has resulted in complex metabolic networks governing metazoan physiology[1,2]. Microbial metabolites influence lipid composition of membranes and fat stores[3–5], modulate signaling pathways that regulate host immune function[6,7], and provide diet-derived metabolic feedback[8,9]. Fatty acid desaturation plays a central role in lipid membrane and metabolic homeostasis[10,11], and is governed by highly conserved stearoyl-CoA desaturase (*SCD1*, Fig. 1A). Correspondingly, perturbation of *SCD1* expression has pervasive effects on physiology, for example, changes

---

[1]Boyce Thompson Institute and Department of Chemistry and Chemical Biology, Cornell University, Ithaca, NY 14853, USA. [2]Chemistry Department, College of Environmental Science and Forestry, State University of New York, Syracuse, NY 13210, USA. [3]Department of Molecular Biology and Genetics, Cornell University, Ithaca, NY 14853, USA. [4]These authors contributed equally: Maximilian J. Helf, Russell N. Burkhardt. ✉e-mail: fs31@cornell.edu

**E** *m/z* 183.1391

**F**

| | Vehicle | 50 μM undecanoic acid | 150 μM undecanoic acid | 50 μM becyp#1 | 150 μM becyp#1 |

P_fat-7_::fat-7::GFP — Brightfield / GFP

**G** Vehicle / 75 μM becyp#1

L4440 (control) — Brightfield / GFP

*nhr-49* RNAi — Brightfield / GFP

in *SCD1* expression are associated with obesity and insulin resistance[12,13] and are a hallmark of inflammatory and autoimmune diseases as well as many cancers[14–17].

Nuclear receptors (NRs) of the liver-X receptor (LXR) and peroxisome-proliferator activated receptor (PPAR) families are master regulators of vertebrate lipid metabolism, including *SCD1* expression (Fig. 1A)[18,19]. Synthetic PPARα agonists, e.g. fibrates, are prescribed to combat metabolic disorders and promote expression of *SCD1*[18,20]. Endogenous regulation of PPARα activity relies on metabolic and nutritional feedback[21–24]; however, it is unclear whether there exist specific, endogenous PPARα agonists that promote *SCD1* expression[25]. Moreover, it is unknown whether microbiota-derived metabolites can regulate fatty acid desaturation at the transcriptional level, despite strong evidence for key roles of

microbiota in host fat metabolism, including fat storage and insulin sensitivity[3–5,26–29].

In *C. elegans*, a tractable model system for lipid metabolism[30] and host-microbe interactions[31], there are three *SCD1* homologs[32], including *fat-7*, which governs the relative abundances of saturated and unsaturated fatty acids[33,34]. *fat-7* expression is primarily regulated via the NR, NHR-49[35], a homolog of human hepatocyte nuclear factor-4α (HNF4α)[36]; however, NHR-49 functionally mimics PPARα, based on similar roles during starvation[37] and the finding that PPARα-targeting fibrates extend *C. elegans* lifespan in an NHR-49-dependent manner[38]. Here we leverage the tractability of *C. elegans* and its dietary bacteria to uncover an ancestral biochemical network that integrates elements of microbial and host metabolism to regulate lipid desaturation via NHR-49/PPARα and FAT-7/SCD1.

**Fig. 1 | β-branched cyclopropyl fatty acids promote desaturation. A** PPARα/ NHR-49 promotes transcription of stearoyl-CoA desaturase enzymes (*SCD1* in humans, *fat-5/6/7* in *C. elegans*), which convert saturated fatty acids (SFAs) to monounsaturated fatty acids (MUFAs) and are implicated in a range of physiologic processes. **B** Schematic summarizing inhibition of *fat-7* expression by ACDH-11, which is proposed to sequester medium chain fatty acids. *fat-7* is highly expressed in *acdh-11* mutants; it was previously hypothesized that this is due to accumulation of medium chain fatty acids that activate NHR-49-dependent transcription of *fat-7*[34]. **C** Maximum likelihood tree for ACDH-11 homologs (see "Methods" for details). ACDH-11 is highly conserved across nematodes in clade V as well as a subset of nematodes (*Strongyloididae*) from clade IV but is distinct from other acyl-CoA dehydrogenase enzymes in *C. elegans*. Bootstrap support values over 50% are indicated. **D** Schematic for comparative metabolomics by HPLC-HRMS of *acdh-11(n5878)* versus the wildtype FAT-7::GFP mixed-stage cultures and volcano plot for subset of features detected in negative ion mode in the *exo*-metabolome. *P*-values were calculated by unpaired, two-sided Welch's *t*-test; no adjustments were made for multiple comparisons, see "Methods" for details. Circled features (blue dashed line) represent octopamine succinyl ascarosides (osas#) that are depleted in *acdh-11(n5878)*[108]. The ω-1 hydroxylated $C_{11}$-cyclopropane fatty acid, named becyp#2, was isolated by chromatographic fractionation and characterized by 2D-NMR. Structural assignment based on $^1H$-$^1H$ couplings marked by green arrows. **E** Synthesis of becyp#1 via Simmons-Smith cyclopropanation and Jones oxidation of *(3Z)-decenol*, see Supporting Information for details. Extracted ion chromatograms (EICs) for *m/z* 183.1391, corresponding to $C_{11}H_{19}O_2^-$, in extracts of N2 or *acdh-11* mutants reared on *E. coli* OP50, or synthetic becyp#1, as indicated. Asterisk (*) marks the *trans*-becyp#1 isomer, an impurity from synthesis. **F** Representative brightfield and GFP fluorescence micrographs of $P_{fat-7}$:*fat-7::GFP* animals reared at 25 °C on *E. coli* JW1653-1 supplemented with vehicle (0.5% ethanol), 50 μM undecanoic acid, 150 μM undecanoic acid, 50 μM becyp#1, or 150 μM becyp#1. Scale bar 0.1 mm. Five independent experiments were performed. **G** Representative brightfield and GFP fluorescence micrographs of $P_{fat-7}$:*fat-7::GFP* animals reared at 25 °C on *E. coli* HT115 expressing L4440 (vector control) or *nhr-49* RNAi and supplemented with vehicle (0.5% ethanol) or 75 μM becyp#1. Scale bar 0.1 mm. Ten independent experiments were performed.

## Results

### A cyclopropyl fatty acid promotes desaturation

In *C. elegans*, the Δ9 desaturases FAT-6 and FAT-7 function as gatekeepers of polyunsaturated fatty acid (PUFA) biosynthesis and convert stearoyl-CoA into oleoyl-CoA (Fig. 1A)[35,39]. Fatty acid desaturation via FAT-7, but not FAT-6, is important for maintaining membrane fluidity during thermal stress. *fat-7* expression is strongly induced by as little as 3 h cold exposure[40] and conversely suppressed during heat exposure[34]. Previous work demonstrated that the acyl-CoA dehydrogenase, ACDH-11, plays a central role in heat adaptation by attenuating NHR-49-dependent expression of *fat-7*[34]. *acdh-11(n5878)* loss-of-function mutants exhibit constitutively high *fat-7* expression, which results in excessive membrane fluidity at elevated temperature (25 °C) causing embryonic lethality and developmental arrest[34]. To explain the heat-sensitive phenotype, it was proposed that ACDH-11 would sequester $C_{10}$-$C_{12}$-straight-chain fatty acids that activate NHR-49, thereby down-regulating expression of *fat-7* (Fig. 1B)[34].

However, phylogenetic analysis indicated that *acdh-11* is distinct from other *C. elegans* acyl-CoA dehydrogenases participating in fatty acid β-oxidation (Fig. 1C), which led us to hypothesize that *acdh-11* may instead be involved in metabolism of a structurally distinct substrate that acts as an NHR-49 agonist. To test this idea, we performed comparative metabolomics of *acdh-11* mutants and WT animals, which revealed accumulation of a large number of previously uncharacterized metabolites in the *acdh-11* mutant (Fig. 1D). To clarify what structural features characterize these *acdh-11*-enriched metabolites, we isolated one of the most abundant compounds via chromatographic fractionation. NMR spectroscopic analysis of the isolated sample revealed a hydroxylated 11-carbon β-cyclopropyl fatty acid (becyp#2, Fig. 1D). Further analysis of molecular formulae and MS2 spectra of the *acdh-11*-enriched metabolites suggested that most of the *acdh-11*-enriched metabolites could plausibly be derived from a corresponding 11-carbon parent β-cyclopropyl fatty acid (βCPFA), named becyp#1 (Fig. 1E). We then confirmed the structure of becyp#1 via chemical synthesis and found that this compound is dramatically enriched in *acdh-11* mutants (Fig. 1E). In fact, comparison of free fatty acid profiles revealed that becyp#1 was the most significantly enriched fatty acid in *acdh-11* mutants relative to WT animals (Supplementary Fig. 1).

becyp#1 is an unusual 11-carbon fatty acid, distinguished from straight-chain fatty acids by a β-cyclopropyl moiety (Fig. 1E), suggesting that its accumulation in *acdh-11* mutants may underlie the dramatically increased *fat-7* expression in this mutant. To test this, we supplemented synthetic samples of becyp#1 and straight-chain undecanoic acid to a transgenic reporter strain that expresses a FAT-7::green fluorescent protein (GFP) fusion driven by the *fat-7* promoter ($P_{fat-7}$:*fat-7::GFP*). Supplementation with becyp#1 strongly induced FAT-7::GFP, whereas induction by straight chain undecanoic acid was weak at the tested concentrations (Fig. 1F). Next, we asked whether FAT-7::GFP induction by becyp#1 requires NHR-49 by using RNAi. Whereas FAT-7::GFP was strongly induced in becyp#1-treated animals reared on *E. coli* HT115 expressing empty vector (L4440), animals reared on HT115 expressing *nhr-49* RNAi did not show any detectable FAT-7::GFP (Fig. 1G). Taken together, our comparative metabolomic analysis revealed a $C_{11}$-β-cyclopropyl NHR-49 agonist, becyp#1, that accumulates in *acdh-11* animals and thus may explain the constitutively high *fat-7* expression in this mutant.

### Bacterial cyclopropyl lipids are the source of becyp#1

Next we investigated the biosynthetic origin of becyp#1. The lipidome of *C. elegans* fed *E. coli*, the most common food source used in the laboratory, contains large amounts of ω−7 $C_{17}$ and $C_{19}$ cyclopropyl lipids, which are derived from bacterial phospholipid membranes[41,42]. We hypothesized that following dietary uptake by *C. elegans*, four rounds of β-oxidation of one of the most abundant bacterial cyclopropane fatty acids, lactobacillic acid (LBA), would produce becyp#1-CoA (Fig. 2A)[42,43]. Due to the presence of the β-cyclopropyl group, becyp#1-CoA would be unsuitable for further processing by canonical β-oxidation enzymes[44], and thus may require a specialized acyl-CoA dehydrogenase such as ACDH-11, explaining accumulation of becyp#1 in *acdh-11* mutants.

To corroborate that becyp#1 and other βCPFAs enriched in *acdh-11* are in fact derived from bacterial cyclopropyl lipids, we performed additional comparative metabolomics of WT *C. elegans* and *acdh-11* mutants reared on *E. coli* BW25113 (WT) or mutant *E. coli* JW1653-1 (Δ*cfa*), a cyclopropane synthase (CFA) knockout strain from the Keio collection that does not produce cyclopropyl lipids (Fig. 2B)[45]. Comparative analysis of *C. elegans* fed WT *E. coli* or Δ*cfa E. coli* revealed dramatic differences. These included several families of metabolites derived from incorporation of bacterial cyclopropane fatty acids into host-dependent lipids, for example series of lysophosphatidylcholines[46] and *N*-acyl glycoglycerophosphoethanolamines[47], which were abundant in *C. elegans* fed WT *E. coli* but abolished in animals fed Δ*cfa E. coli* (Supplementary Fig. 2). Significantly, almost all of the metabolites enriched in *acdh-11* mutants grown on WT bacteria, including becyp#1 (Fig. 2C) and other metabolites we had previously shown to be derived from cyclopropyl lipids[47], were absent from the metabolomes of WT *C. elegans* and *acdh-11* animals reared on Δ*cfa E. coli* (Supplementary Table 1), confirming that becyp#1 is derived from bacterial cyclopropyl lipids. In addition, our analyses showed that fatty acid profiles of both WT *C. elegans* and *acdh-11* mutants are broadly altered when fed Δ*cfa E. coli*. For example, monounsaturated vaccenic acid and palmitoleic acid and a subset of polyunsaturated fatty acids were increased in both WT animals and *acdh-11* mutants reared on Δ*cfa E. coli* (Supplementary

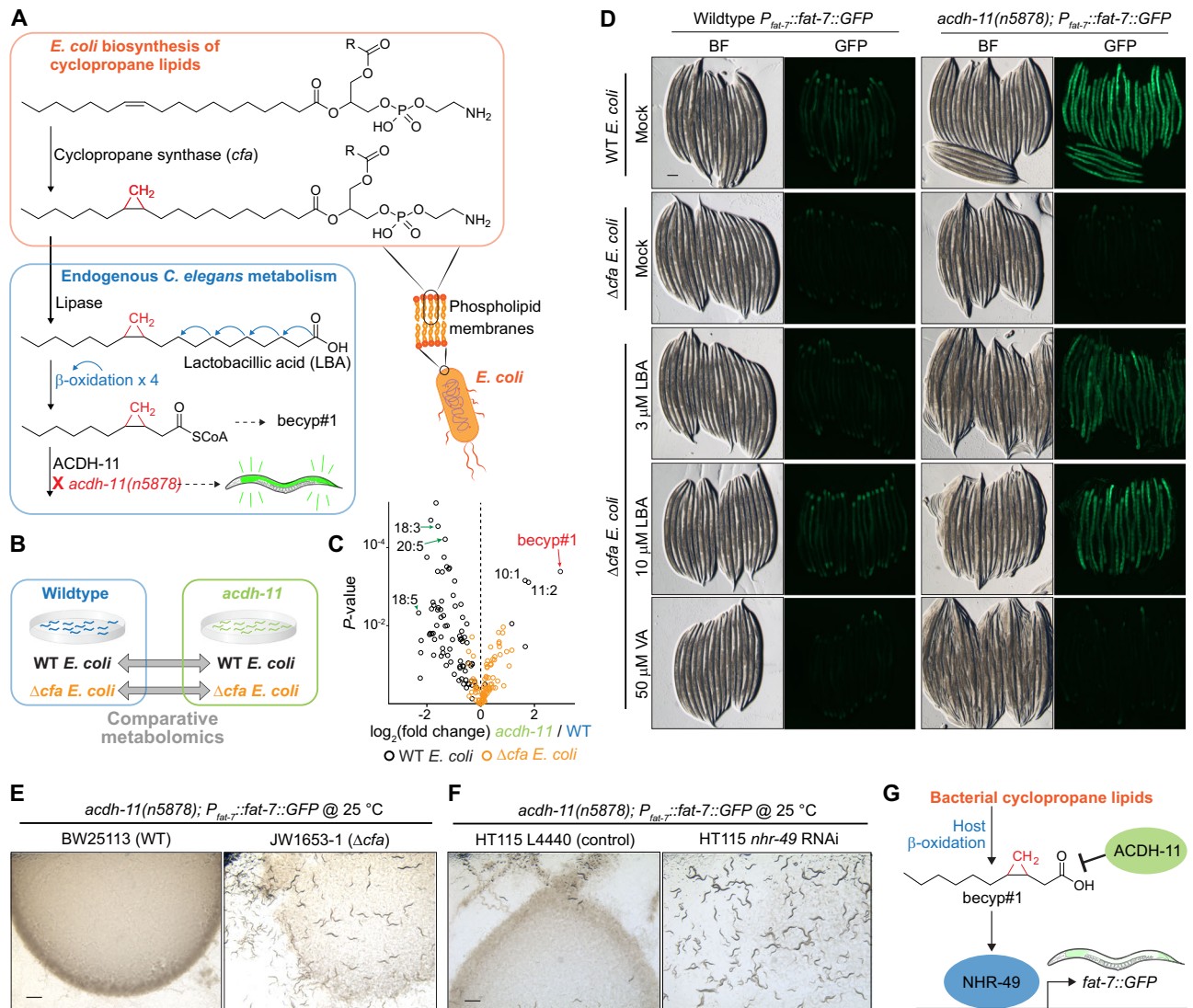

**Fig. 2 | Bacterial cyclopropyl lipids are the source of becyp#1. A** Cyclopropyl lipids are synthesized by the *E. coli* cyclopropyl fatty acid synthase (CFA), which converts vaccenic acid or palmitoleic acid to the corresponding cyclopropyl derivatives in the context of phospholipids, in which R represents an additional acyl group of the phospholipid[42]. Lipolysis and β-oxidation of CPFAs by *C. elegans* produces a $C_{11}$ β-cyclopropyl fatty acid, becyp#1, that accumulates in *acdh-11* loss-of-function mutants. **B** Schematic for comparative metabolomics of *acdh-11(n5878)* versus wildtype FAT-7::GFP fed either WT *E. coli* or Δ*cfa E. coli*. **C** Overlaid volcano plots for 85 free fatty acids detected by HPLC-HRMS (negative ion, post-column ion pairing) in the *endo*-metabolomes of WT and *acdh-11(n5878)* mutant *C. elegans* fed WT *E. coli* (black dots) or Δ*cfa E. coli* (orange dots). *P*-values were calculated by unpaired, two-sided Welch's *t*-test; no adjustments were made for multiple comparisons. The most significantly enriched free fatty acid in *acdh-11* mutants fed WT *E. coli* is becyp#1; additional cyclopropane-containing medium chain fatty acids are also enriched in *acdh-11* mutants relative to WT *C. elegans*. The free fatty acid

profiles of WT and *acdh-11* animals are similar when fed Δ*cfa E. coli*. Volcano plots focused on dietary differences within each genotype are displayed in Supplementary Fig. 2G. **D** Representative brightfield (BF) and GFP fluorescence micrographs of WT and *acdh-11(n5878)* $P_{fat-7}$::*fat-7*::GFP animals reared at 20 °C on WT *E. coli*, Δ*cfa E. coli*, or Δ*cfa E. coli* supplemented with lactobacillic (LBA) or vaccenic acid (VA), as indicated. Scale bar 0.1 mm. Three independent experiments were performed. **E** Representative brightfield micrographs of *acdh-11* mutants reared on BW25113 (WT) or JW1653-1 (Δ*cfa*) *E. coli* at 25 °C. Twelve L1 larvae were seeded onto each plate and photographed after five days. Scale bar 1 mm. Six independent experiments were performed. **F** Representative brightfield micrographs of *acdh-11* mutants reared on HT115 *E. coli* expressing L4440 (vector control) or *nhr-49* RNAi at 25 °C. Twenty L1 larvae were seeded onto each plate and photographed after four days. Scale bar 1 mm. Six independent experiments were performed. **G** Revised model of ACDH-11 regulation of *fat-7* expression. Bacteria-derived βCPFAs accumulate in the *acdh-11(n5878)* mutant and drive NHR-49-dependent *fat-7* expression.

---

Fig. 2G), consistent with a previous report[48]. Vaccenic and palmitoleic acids are increased in animals reared on Δ*cfa E. coli* likely because these two fatty acids are the substrates for *E. coli* cyclopropane synthase[42] and thus accumulate in the absence of the CFA enzyme.

Next, we investigated the role of bacterial cyclopropyl fatty acid production for *acdh-11*-dependent phenotypes. Consistent with previous studies[34], strong FAT-7::GFP expression was observed at all life stages in *acdh-11* mutants reared on WT *E. coli* diets, including in unhatched eggs (Fig. 2D, Supplementary Fig. 3). However, FAT-7::GFP expression was sharply reduced in *acdh-11* animals reared on Δ*cfa E.*

*coli* as compared to animals reared on WT *E. coli* (Fig. 2D). Supplementation of *acdh-11* animals reared on Δ*cfa E. coli* with the bacterial cyclopropane fatty acid, LBA, resulted in dose-dependent recovery of FAT-7::GFP expression (Fig. 2D), along with concomitant recovery of βCPFA production (Supplementary Fig. 4). To a lesser extent, LBA supplementation also induced FAT-7::GFP in WT animals (Fig. 2D). In contrast, supplementation with vaccenic acid, the monounsaturated precursor of LBA we used as a control, had no effect on FAT-7::GFP expression in *acdh-11* mutants or WT animals, even at much higher concentrations (Fig. 2D). Interestingly, FAT-7::GFP expression in WT

animals is most intense in the proximal and distal ends of the intestines, whereas FAT-7::GFP is uniformly strong throughout the intestinal tract of *acdh-11* mutants, possibly due to intestinal build-up of becyp#1 (Fig. 2D).

Given the dramatic reduction in FAT-7::GFP expression in *acdh-11* mutants fed Δ*cfa E. coli*, we next investigated the effect of bacterial cyclopropyl lipid production on viability of *acdh-11* mutants at elevated temperatures. As previously reported, loss of *acdh-11* function results in *fat-7*-dependent embryonic lethality and developmental arrest at 25 °C[34]. Strikingly, embryonic lethality at 25 °C was abolished when rearing *acdh-11* mutant animals on cyclopropyl fatty acid-deficient Δ*cfa E. coli* (Fig. 2E). In addition, *acdh-11* mutants hatched and developed normally at 25 °C when fed cyclopropyl-containing HT115 expressing *nhr-49* RNAi, demonstrating that βCPFAs are not inherently toxic (Fig. 2F). Collectively, these results establish that the *acdh-11* heat-sensitive phenotype is the result of NHR-49-dependent hyperactivation of *fat-7* expression by becyp#1, which is derived from bacterial cyclopropyl lipids and accumulates in *acdh-11* mutants (Fig. 2G).

## An endogenous activator of *fat-7* expression

While the identification of becyp#1 explains *fat-7* hyperactivation in *acdh-11* mutants under laboratory conditions, it seemed unlikely that control of *fat-7* expression via *nhr-49* would rely entirely on one specific type of bacterial lipid. Desaturase expression is controlled by a regulatory network that integrates nutrient availability, pathogen responses, and reproductive signals[49–52]. This suggested the existence of additional small molecule agonists of NHR-49, possibly a metabolite that is structurally related to becyp#1. Surveying existing data for the *C. elegans* metabolome, we noted that mutants defective in *hacl-1* (2-hydroxy-acyl-CoA lyase 1), a key component of conserved peroxisomal fatty acid α-oxidation, accumulate large quantities of 11-carbon β-methyl fatty acids (βMFAs), e.g., bemeth#3[47], whose structures are reminiscent of the 11-carbon βCPFAs we identified as shunt metabolites in *acdh-11* mutants (Fig. 3A).

The most abundant βMFAs identified from *hacl-1* mutants are α-hydroxylated, suggesting that they represent shunt metabolites derived from incomplete α-oxidation of a β-methyldecenoic acid precursor via hydroxylation by the phytanoyl-CoA dioxygenase homologs ZK550.5 and/or ZK550.6 (Fig. 3A). Inspection of the free fatty acid profile of wildtype animals in fact revealed the presence of β-methyldecenoic acid (bemeth#1), as confirmed using a synthetic standard (Fig. 3B). Considering the striking structural similarity of bemeth#1 with the bacterial-derived becyp#1, we hypothesized that bemeth#1 may represent an endogenous regulator of *nhr-49*-dependent *fat-7* expression. Like becyp#1, bemeth#1 is an 11-carbon fatty acid featuring an unusual β-branched carbon skeleton, distinguished by a β-methyl group instead of the β-cyclopropyl ring as in becyp#1. Therefore, we tested synthetic bemeth#1 and becyp#1 in parallel for activation of FAT-7::GFP expression. We found that, similar to becyp#1, supplementation with synthetic bemeth#1 strongly induced FAT-7::GFP (Fig. 3C). Supplementation with the α-hydroxylated derivative, bemeth#2, did not induce FAT-7::GFP, nor was expression of FAT-7::GFP increased in *hacl-1(tm6725)* mutants in which α-hydroxylated βMFAs accumulate (Supplementary Fig. 5)[47].

Because metabolite-receptor interactions are often highly stereospecific[53,54], we developed a synthesis employing Sharpless chiral resolution and subsequent chirality transfer via Claisen rearrangement to access the two enantiomers of bemeth#1, *(3 R)*-bemeth#1 and *(3 S)*-bemeth#1 (Fig. 3D, see "Supplementary Methods" for details)[55]. We next determined the absolute configuration of natural bemeth#1 and bemeth#2 using a chiral oxidation and derivatization approach, which revealed *(3 R)*-bemeth#1 and *(2 R,3 S)*-bemeth#2 as the predominant natural isomers (Fig. 3D, see "Supplementary Methods" for details). The *(2 R,3 S)*-configuration of bemeth#2, indicative of *syn*-hydroxylation, is consistent with the predicted specificity of the *C.*

*elegans* α-oxidation enzymes ZK550.5 and ZK550.6, based on their homology to human PHYH[56].

Testing synthetic samples of the two enantiomers of bemeth#1, we found that low micromolar concentrations of *(3 R)*-bemeth#1 strongly induced FAT-7::GFP, whereas treatment with *(3 S)*-bemeth#1 resulted in significantly lower induction (Fig. 3E, F). The residual activity of the *(3 S)*-bemeth#1 sample used in this study may be attributable to contamination with about 15% of the other enantiomer, resulting from the limited selectivity of available synthetic routes for chiral methyl-branched fatty acids[57]. We also tested for effects of bemeth#1 supplementation on expression of *fat-6*, a desaturase that is functionally redundant with FAT-7 but regulated by different mechanisms[35]. Examination of a FAT-6::GFP transgenic reporter strain showed that expression is unaffected by bemeth#1 (Supplementary Fig. 6). Finally, we used RT-PCR to measure *fat-6* and *fat-7* expression of non-transgenic WT animals supplemented with bemeth#1 and observed a greater than tenfold increase in *fat-7* expression, whereas expression of *fat-6* was unchanged, consistent with the results for the transgenic reporters (Fig. 3G). Taken together, these results demonstrate that, like becyp#1, bemeth#1 regulates *fat-7* expression.

## A methyltransferase conserved across Nematoda

β-branched fatty acids are highly unusual because they cannot be obtained via canonical fatty acid biosynthesis from acetate or propionate, nor from leucine-derived monomethyl branched chain fatty acids which contain a methyl branch at the terminal carbon (Fig. 4A)[39,58,59]. We therefore hypothesized that the unusual β-methyl group is installed by an *S*-adenosyl methionine (SAM)-dependent methyltransferase. To test this, we used a stable isotope labeling approach in which *hacl-1* larvae were supplemented with $D_3$-methyl methionine ($D_3$-Met) in the absence of bacteria. Analysis of $D_3$-Met-supplemented *hacl-1* larvae by HPLC-HRMS revealed incorporation of three deuterium atoms in all annotated βMFAs, including the putative parent compound bemeth#1 and the α-hydroxylated derivatives bemeth#2 and bemeth#3 (Fig. 4B, Supplementary Fig. 7). These results support the idea that βMFA biosynthesis is not dependent on bacteria, but rather requires an endogenous SAM-dependent methyltransferase.

In *C. elegans*, as in humans, the overwhelming majority of SAM-dependent methyltransferases catalyze methylation of nitrogen or oxygen, e.g., *N*-methylation of nucleosides or amino acids[60]. Of the 121 annotated SAM-dependent methyltransferases in *C. elegans*, there are only seven enzymes that are predicted to methylate carbon (*C*-methylation): these include sterol methyltransferase (*strm-1*)[61], ubiquinone *C*-methyltransferase (*coq-5*)[62], and four annotated cytosine $C^5$-methyltransferases[63]. The one remaining predicted *C*-methyltransferase, *F13D12.9*, is highly conserved across Nematoda and exhibits notable similarity to bacterial cyclopropane synthases. Phylogenetic analysis revealed that the closest homologs of *F13D12.9* are found in nematodes and bacteria, not in other animals, consistent with ancient, horizontal gene transfer (HGT) (Fig. 4C)[64]. Correspondingly, *F13D12.9* is annotated as a mycolic acid cyclopropane synthase; however, we and others have shown that *C. elegans* does not synthesize cyclopropane lipids, suggesting that *F13D12.9* could instead be involved in βMFA biosynthesis[41,43,47].

To test the role of *F13D12.9* in the biosynthesis of βMFAs, we obtained two predicted loss-of-function alleles; *F13D12.9(gk155709)* containing an early stop codon (K127*), and *F13D12.9(tm2382)* with a deletion spanning exons 2-4 (Fig. 4D). Comparative metabolomic analysis of wildtype and the two *F13D12.9* mutants revealed that *F13D12.9* is strictly required for production of bemeth#1 and other detected βMFA-derived metabolites (Fig. 4E, summarized in Supplementary Table 2). We therefore named *F13D12.9* as Fatty acid *C*-Methyl Transferase (*fcmt-1*). Consistent with conservation of *fcmt-1* in other nematodes, βMFAs were also detected in the closely related *C. briggsae* (Supplementary Fig. 8)[65].

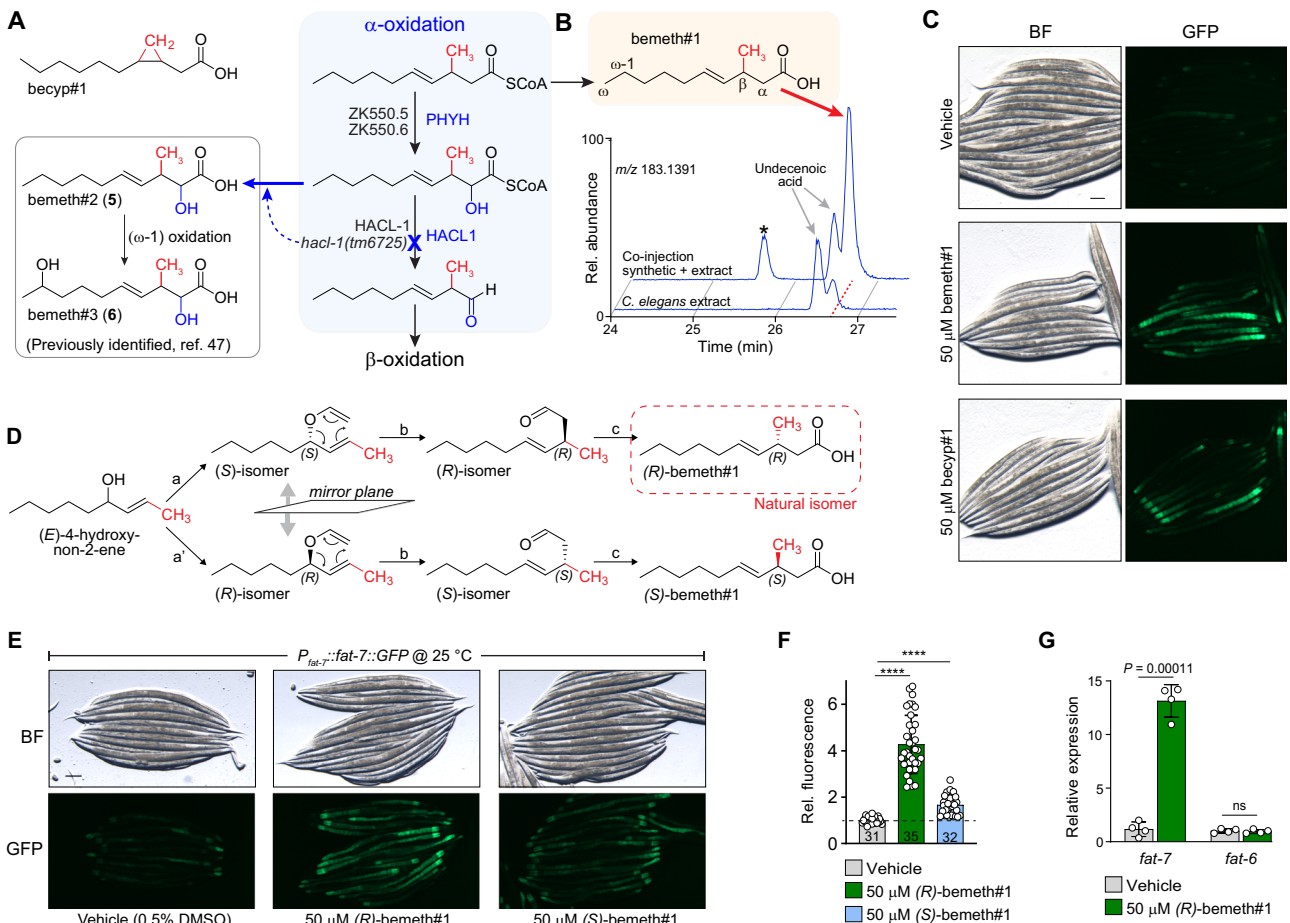

**Fig. 3 | An endogenous activator of *fat-7* expression. A** β-cyclopropyl fatty acids, such as becyp#1, are reminiscent of β-methyl fatty acids (βMFAs) that accumulate in the *hacl-1(tm6725)* mutant, e.g., bemeth#2 and bemeth#3, which were previously characterized[47]. Predicted biosynthesis of βMFAs begins with α-hydroxylation of a β-methyl decenoic acid precursor via one or both of the *C. elegans* homologs of PHYH (ZK550.5, ZK550.6). HACL-1 catalyzes the C-C bond breaking step that yields an aldehyde, which is likely oxidized to an α-methyl fatty acid that can undergo β-oxidation. *C. elegans* enzymes are in black font and the corresponding human homologs are in blue font. **B** EICs for *m/z* 183.1391, corresponding to $C_{11}H_{19}O_2^-$, from *endo*-metabolome extract of N2 larvae, and a co-injection of 500 nM synthetic bemeth#1 (3-methyl-4-*E*-decenoic acid) in extract of N2 larvae, as indicated. Red dashed line highlights bemeth#1, which elutes slightly later than an unknown structural isomer ("undecenoic acid"). Asterisk (*) marks 3-methyl-4-*Z*-decenoic acid, "*cis*-bemeth#1," an impurity from synthesis. No *cis*-bemeth#1 was detected in any biological samples. **C** Representative brightfield (BF) and GFP fluorescence micrographs of $P_{fat-7}$:*fat-7*::*GFP* animals reared at 25 °C on BW25113 (WT) *E. coli* and supplemented with vehicle (0.5% ethanol), 50 µM bemeth#1, or 50 µM becyp#1, as indicated. Scale

bar 0.1 mm. Four independent experiments were performed. **D** Overview of syntheses to afford enantiomerically enriched (*R*)- or (*S*)-bemeth#1. a, a'. Sharpless chiral resolution and conversion to vinyl ethers; b. Claisen rearrangement to branched aldehyde; c. chlorite oxidation. Absolute configuration of natural bemeth#1 was determined via Mosher derivatization, See "Supporting Information" for details. **E, F** Representative brightfield (BF) and GFP fluorescence micrographs (**E**) and quantification of fluorescence intensity (**F**) of $P_{fat-7}$:*fat-7*::*GFP* animals reared at 25 °C supplemented with vehicle (0.5% DMSO or ethanol), 50 µM (*R*)-bemeth#1, or 50 µM (*S*)-bemeth#1. Scale bar 0.1 mm. Data represent four independent experiments in which 6–12 animals per treatment were quantified, bars indicate mean ± s.d. ****$P < 0.0001$ calculated by one-sided Welch's ANOVA with post-hoc Dunnett's T3 multiple comparisons test, $n = 31$ vehicle, $n = 35$ (*R*)-bemeth#1, and $n = 32$ (*S*)-bemeth#1. **G** Relative expression of *fat-6* and *fat-7* desaturase genes in N2 (WT) animals supplemented with vehicle (0.5% DMSO) or 50 µM (*R*)-bemeth#1 as determined by RT-PCR, normalized to *act-1*. Data represent four independent experiments and error bars indicate mean ± s.d. *P* values calculated using unpaired, two-sided *t*-test with Welch correction. ns, not significant. Source data are provided as a Source Data file.

## NHR-49 agonists integrate bacterial and endogenous metabolism

Our metabolomic analyses revealed two remarkably similar NHR-49 agonists, becyp#1 and bemeth#1, that are derived from parallel biosynthetic pathways involving homologous bacterial and endogenous enzymes, cyclopropane fatty acid synthase (CFA) in *E. coli* and FCMT-1 in *C. elegans*, which likely originated from bacterial CFA via HGT (Fig. 4C). In *E. coli*, CFA acts upon phospholipids containing singly unsaturated *cis*-16:1 (palmitoleic) or *cis*-18:1 (vaccenic) ω-7 fatty acids (Fig. 2A)[41,42]. CFA homologs in other bacterial species produce diverse cyclopropyl and methyl-branched lipids, e.g., the mycolic acids of *Mycobacterium tuberculosis*[66].

To clarify the parallel roles of bacterial CFA and endogenous FCMT-1 for the synthesis of becyp#1 and bemeth#1, respectively, we

considered possible substrates for FCMT-1. Based on its homology to bacterial CFAs, we hypothesized that FCMT-1 should catalyze methyl transfer to an ω-7 *cis*-double bond, as in *cis*-VA, which is abundant in *E. coli*[67]. Intriguingly, the *Rhodobacter sphaeroides* CFA homolog, UfaM (FCMT-1 BLAST E-value 5e-72), catalyzes methyl transfer to VA with concomitant double bond migration, resulting in an 11-methyl-12-*trans* 18:1 fatty acid, a plausible precursor of bemeth#1 (Fig. 5A)[68]. Therefore, we performed stable isotope feeding experiments using $D_{13}$-*cis*-VA (Fig. 5B). As an additional control, we fed $D_{13}$-*trans*-VA, reasoning that, based on homology, the *trans* double bond may not function as a methyl acceptor. Analysis of $D_{13}$-*cis*- and $D_{13}$-*trans*-VA-supplemented animals by HPLC-HRMS showed that all *fcmt-1*-dependent metabolites incorporate deuterium label from $D_{13}$-*cis*- but not $D_{13}$-*trans*-VA (Fig. 5C). These results indicate that FCMT-1 acts on *cis*-double bonds

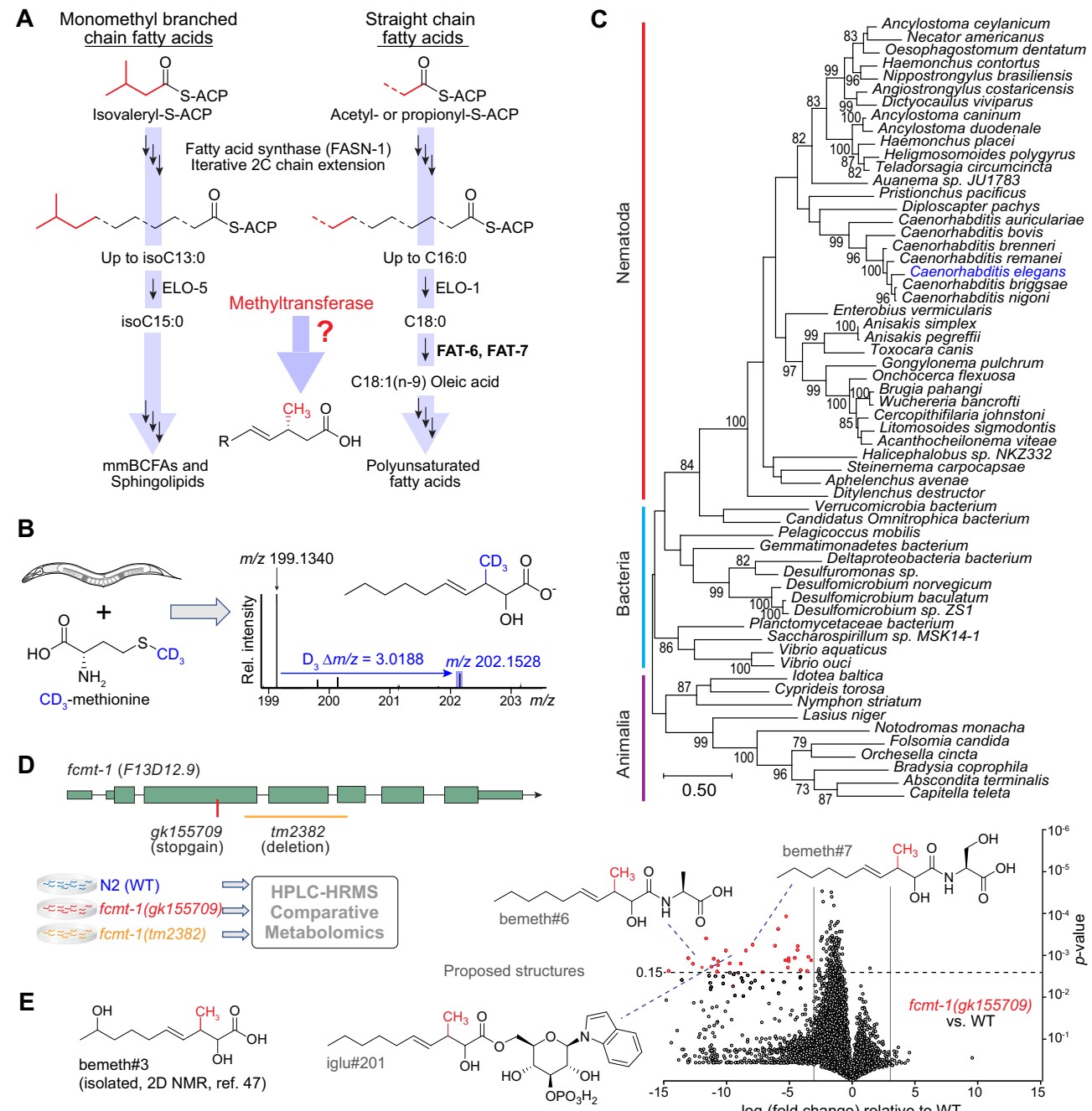

**Fig. 4 | A methyltransferase conserved across Nematoda. A** Canonical straight chain fatty acid biosynthesis in *C. elegans* begins with either acetyl- or propionyl-CoA and proceeds via iterative 2-carbon condensation cycles to produce even- or odd-numbered fatty acids up to 16 carbons in length. Biosynthesis of monomethyl branched chain fatty acids begins with isovaleryl-CoA derived from Leu, in which the methyl branch is located at the terminal (ω) carbon. No known biosynthetic pathways exist for the production of β-methyl branched fatty acids, suggesting the activity of a methyltransferase. ACP, acyl carrier protein; R, remainder of bemeth#1. **B** *hacl-1(tm6725)* larvae were supplemented with 5 mM D₃-methyl methionine (CD₃-Met) in the absence of bacteria, and extracts were analyzed by HPLC-HRMS. Mass spectrum for bemeth#2 exhibits an isotope with an exact *m/z* shift of 3.0188 (blue), corresponding to incorporation of exactly three deuterium atoms, thereby implicating an endogenous methyltransferase. **C** Maximum likelihood tree for FCMT-1 homologs. Sequences of the top 50 Blastp hits in the NCBI-nr database were aligned along with the top 10 Blastp hits in animals (excluding nematodes). The closest homologs are found in nematodes and bacteria, not in other animals, pointing to a

possible horizontal gene transfer (HGT) event in a Nematode ancestor[64]. Bootstrap support values over 70% are indicated in the tree. **D** Schematic of *fcmt-1* (previously *F13D12.9*) gene structure and comparative metabolomics. *fcmt-1(gk155709)* harbors a nonsense mutation resulting in an early stop codon, K127*, and *fcmt-1(tm2382)* harbors a genomic deletion spanning exons 2–4. Small olive rectangles represent untranslated regions, large rectangles represent exons, and black lines represent introns. The indicated strains were grown and extracted in parallel, then analyzed by HPLC-HRMS using Metaboseek software for comparative metabolomics. **E** Volcano plot for subset of features detected by HPLC-HRMS (negative ion) in the *exo*-metabolomes of *fcmt-1(gk155709)* versus the wildtype (N2) control. *P*-values were calculated by unpaired, two-sided Welch's *t*-test; dashed line represents Benjamini-Hochberg adjusted significance threshold at 15% false discovery rate, see "Methods" for more details. bemeth#3 was previously characterized[47]; additional proposed structures based on MS/MS fragmentation and stable isotope enrichment, see Supplementary Table 2 for a complete list of *fcmt-1*-dependent features.

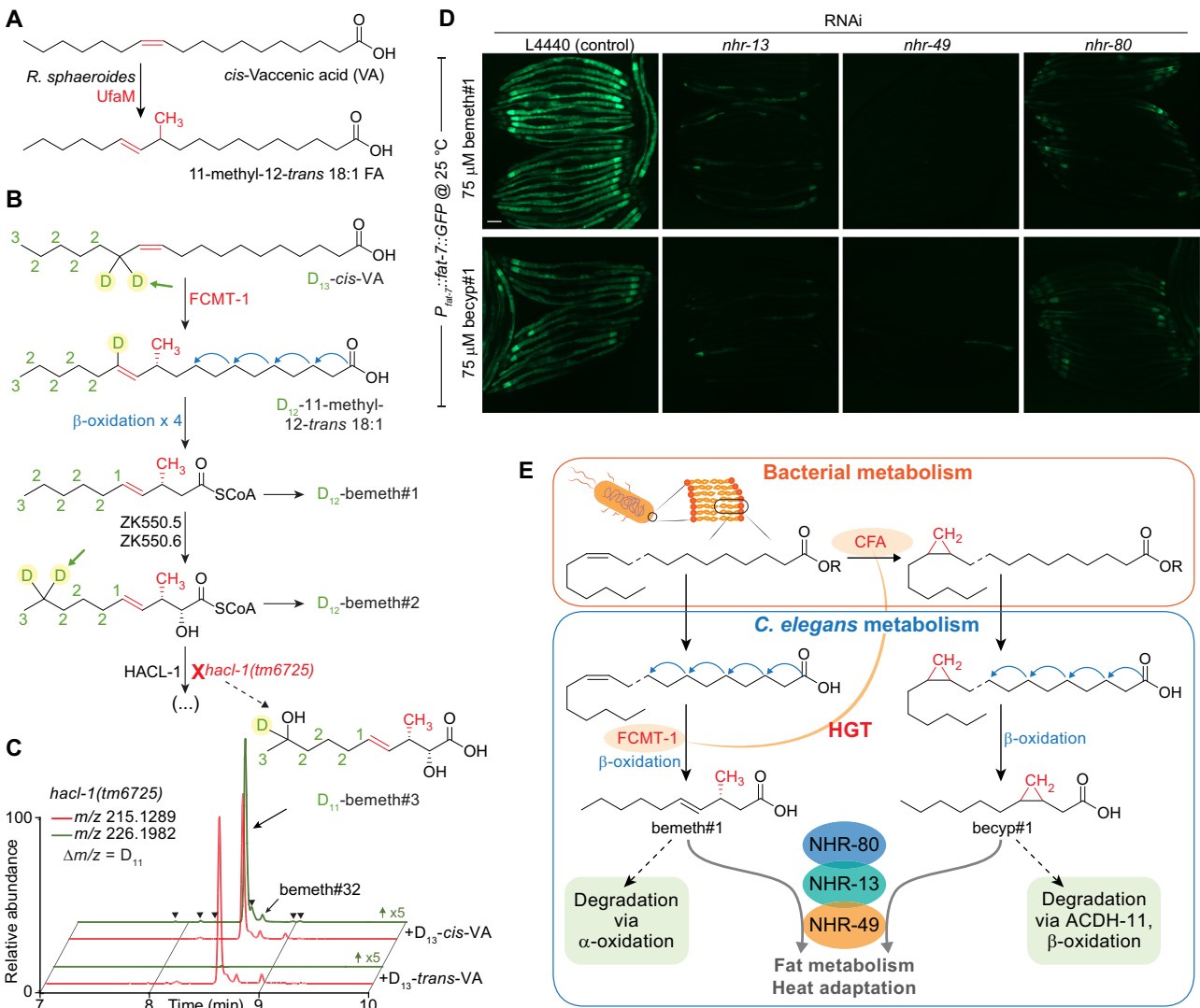

**Fig. 5 | NHR-49 agonists integrate bacterial and endogenous metabolism. A** The *Rhodobacter sphaeroides* SAM-dependent methyltransferase, UfaM (FCMT-1 BLAST E-value 5e[68]), catalyzes methyl transfer to VA to produce an 11-methyl-12-*trans* 18:1 FA[68]. **B** Proposed biosynthesis of bemeth#1 based on isotopically labeled D13-vaccenic acid (VA) feeding experiment (number of D atoms denoted in green next to each carbon). Methyl transfer results in the abstraction of one deuterium atom (highlighted with green arrow) and distal oxidation results in metabolites with a diagnostic number of deuterium atoms remaining. **C** EICs for *m/z* 215.1289 and 226.1982, corresponding to bemeth#3 and D11-bemeth#3, respectively, in extracts of *hacl-1* cultures supplemented with D13-*cis*- or D13-*trans*-VA. Isotopically enriched bemeth#3 is observed when *hacl-1* animals are supplemented with D13-*cis*- but not D13-*trans*-VA. Y-axis for *m/z* 226.1982 is scaled 5-fold to clearly show traces for labelled features. Isotope enrichment in low-abundance isomers confirmed the existence of several additional bemeth#3 isomers (bemeth#33 – bemeth#38,

marked by arrowheads, see Supplementary Table 2). **D** Representative GFP fluorescence micrographs of $P_{fat-7}$::*fat-7::GFP* animals reared at 25 °C on HT115 *E. coli* expressing various RNAi and supplemented with either 75 μM bemeth#1 or 75 μM becyp#1, as indicated. The scale bar represents 0.1 mm. Five independent experiments were performed. **E** Top: Cyclopropyl lipids are synthesized by the *E. coli* cyclopropyl fatty acid synthase, CFA, which converts VA or palmitoleic acid to the corresponding cyclopropyl derivatives in membrane phospholipids. R represents remainder of phospholipid. Bottom: VA is the substrate for FCMT-1 methylation; four rounds of β-oxidation produce bemeth#1, a *fat-7* agonist that requires NHR-13, −49, and −80. Degradation of bemeth#1 occurs via α-oxidation, which yields an α-methyl aldehyde compatible with β-oxidation. In parallel, lipolysis and β-oxidation of bacterial CPFAs by *C. elegans* produces a C11 β-cyclopropyl fatty acid, becyp#1, which signals via overlapping NHRs. becyp#1 is likely degraded by ACDH-11-dependent β-oxidation, which is predicted to yield an α-methyl aldehyde.

of monounsaturated fatty acids, in close analogy to bacterial enzymes such as UfaM. Moreover, partial loss of the deuterium label at specific positions in the *fcmt-1*-dependent metabolites from D13-*cis*-VA-supplemented animals, e.g., at position 5 in bemeth#1 (Fig. 5B, arrow), provided additional corroboration of the proposed biosynthetic pathway (Supplementary Fig. 9). Taken together, our results indicate that the two NHR-49 agonists, becyp#1 and bemeth#1, are produced via parallel pathways from ω-7 fatty acids common in *E. coli*. The βMFA bemeth#1 is produced via a fully endogenous pathway, in which VA is likely *C*-methylated by FCMT-1 and chain-shortened via canonical β-oxidation, whereas becyp#1 is derived from conversion

of VA to LBA by bacterial CFA, followed again by endogenous β-oxidation.

Given their structural similarity, we asked whether there are any overt differences in the activation of *fat-7* expression by bemeth#1 and becyp#1. NHR-13, NHR-80 and NHR-66 have been characterized as direct NHR-49-interactors[69], whereby NHR-13 and NHR-80 affect expression of desaturase genes, including *fat-7*, whereas NHR-66 cooperates with NHR-49 to repress genes related to sphingolipid and phospholipid metabolism[69,70]. Induction of FAT-7::GFP expression by both bemeth#1 and becyp#1 was reduced, but not abolished, in animals reared on *nhr-80* or *nhr-13* RNAi (Fig. 5D). In animals reared on

*nhr-80* RNAi, FAT-7::GFP was expressed weakly in the first intestinal cell and additionally around the pharynx, as well as in the extreme posterior cells (Fig. 5D). A subset of animals reared on *nhr-13* RNAi exhibited mosaic FAT-7::GFP in response to supplemented bemeth#1 or becyp#1, in which expression was restricted to the tail and posterior intestinal cells (Fig. 5D). Animals reared on RNAi against *nhr-66* did not exhibit altered FAT-7::GFP expression in response to supplemented fatty acids, nor did animals fed *hlh-30* RNAi, which cooperates with *nhr-49* in starvation responses (Supplementary Fig. 10)[69,71]. Thus, supplementation with either bemeth#1 or becyp#1 increased *fat-7* expression in a manner that was strictly dependent on NHR-49 and partially dependent on NHR-13 and NHR-80, consistent with a model in which bemeth#1 and becyp#1 modulate fat metabolism via tissue-specific heterodimers of NHR-49 with NHR-13 and NHR-80[69].

## Discussion

In this work, we identified parallel biosynthetic pathways that control fatty acid desaturation via NHR-49/PPARα, (i) a pathway dependent on microbiota-derived cyclopropane lipids, which are converted endogenously into the β-cyclopropane 11-carbon fatty acid, becyp#1, and (ii) a fully endogenous pathway that requires the HGT-derived methyltransferase FCMT-1 for biosynthesis of the β-methyl-branched 11-carbon fatty acid, bemeth#1 (Fig. 5E). Both becyp#1 and bemeth#1 strongly promote expression of the desaturase FAT-7 in a manner that requires NHR-49 and is partially dependent on NHR-13 and NHR-80. Despite their structural similarity, becyp#1 degradation proceeds via a dedicated β-oxidation enzyme, ACDH-11, whereas bemeth#1 is degraded via α-oxidation, analogous to human metabolism of β-methyl fatty acids, e.g., dairy-derived phytanic acid[47,72]. Their distinct degradation pathways suggest that *fat-7* activation by microbiota-dependent becyp#1 and endogenous bemeth#1 is regulated via separate mechanisms, which may be required to accommodate fluctuating nutritional and environmental conditions.

Microbiota-derived metabolites are deeply embedded in the fabric of metazoan physiology[1]. Examples include the microbial metabolism of bile acids[73,74], which function in nutrient absorption and as endogenous agonists of farnesoid-X receptor (FXR)[75,76], as well as propionate and butyrate[77], simple short-chain fatty acids that play key roles in the maintenance of intestinal immune homeostasis[78]. The example of becyp#1 demonstrates that microbiota-specific lipids, i.e., cyclopropane fatty acids, can also function as precursors for potent signaling molecules. Metagenomic analyses of the human microbiome indicate that the most abundant phyla of human-associated bacteria harbor genes encoding cyclopropane synthase[79], although the extent to which microbes produce cyclopropane lipids in the gut has not been investigated. That cyclopropane lipids may act as specific signaling molecules in humans is suggested by in vitro assays in which cyclopropane fatty acids, but not simple saturated fatty acids, activated the G-protein coupled receptors brain angiogenesis factor 1 (BAI1)[80] and G-protein coupled receptor 84 (GPR84)[81]. Immune responses elicited by specific methyl-branched phospholipids produced by the gut bacteria *Akkermansia muciniphila* provide another example demonstrating that small structural differences of fatty acids can dramatically affect biological responses[82].

In addition to the gut microbiome, cyclopropyl lipids are present in human diets, and have recently become more abundant in meat and dairy products as a result of the widespread introduction of fermented grains (silage) as animal feed during the 20th century[83]. The grain ensiling process involves bacterial fermentation[84], and the bacteria-derived cyclopropyl lipids are then passed up the food chain to humans. While little is known about the metabolic fate of cyclopropyl lipids in humans, cyclopropyl fatty acids are robustly detected in human serum, and may interact directly with the human NR, HNF4α, a master regulator of mammalian liver metabolism and homolog of NHR-13, -49, and -80[85,86]. Early biochemical studies showed that

cyclopropyl lipids are highly persistent in rats, accumulating as shorter-chain derivatives in adipose tissue[87,88], and elevated levels of circulating cyclopropane fatty acids have been associated with inflammatory disease[89]. Taken together, these preliminary observations and our finding that cyclopropane fatty acids function as potent signaling molecules in *C. elegans* may motivate further studies of the physiological roles of cyclopropyl lipids in humans[90].

In contrast to mammals, *C. elegans* appears able to degrade cyclopropyl lipids via ACDH-11, a specialized β-oxidation enzyme that is not conserved in humans. Expression of *acdh-11* prevents hyper-activation of NHR-49 due to accumulation of becyp#1, possibly representing a dietary adaptation since the genomes of many naturally *C. elegans*-associated bacteria include predicted cyclopropane synthases[91]. Consistent with a direct functional role in homeoviscous adaptation, *acdh-11* is heat-inducible[40], resulting in increased becyp#1 degradation at elevated temperature, which reduces NHR-49- and FAT-7-dependent fatty acid desaturation, thereby reducing membrane fluidity and thus promoting heat adaptation. Because cyclopropane biosynthesis varies between different bacteria and can be stress- or growth phase-dependent[92–94], the extent to which cyclopropyl lipids impact *C. elegans* physiology may vary considerably in different natural and laboratory environments. Correspondingly, rearing *acdh-11* mutants on *E. coli* lacking the cyclopropane synthase gene (Δ*cfa*) fully rescues their heat-sensitive phenotype. *acdh-11* homologs are highly conserved in many other bacterivorous and also parasitic nematodes (Fig. 1C), suggesting that the role of *acdh-11* homologs in the sequestration of β-cyclopropyl fatty acids may also be conserved.

Conservation of the methyltransferase *fcmt-1* across Nematoda is consistent with an ancient HGT event that conferred to nematodes the endogenous capacity to produce otherwise inaccessible β-methyl-branched fatty acids. We showed that production of 11-carbon β-methyl fatty acids is conserved in *C. briggsae*, suggesting that β-methyl fatty acids function as NHR-49/PPARα agonists also in other nematode species. We infer that there is considerable flux through the bemeth#1 pathway based on the dramatic accumulation of oxidized βMFAs in starved *hacl-1(tm6725)* larvae[47]. Life stage- and sex-specific functions of FCMT-1 are further suggested by the recent observations that production of specific bemeth#-family metabolites is greatly increased during the transition from late larval (L4) stage to reproductive adult[47], and is increased further in adult males[95], consistent with publicly available transcriptomic data suggesting that *fcmt-1* is expressed in the male germline[96,97]. Further investigation into the life stage- and tissue-specific regulation of NHR-49 by β-branched fatty acids may reveal additional evolutionary insight into the acquisition of a bacterial methyltransferase by an ancient metazoan.

Collectively, our results showcase a central role for microbiota in the regulation of metazoan fatty acid metabolism by demonstrating that a microbiota-derived cyclopropyl fatty acid can activate NR-dependent desaturase expression. Further, our findings suggest an ancient origin for microbial regulation of nematode fat metabolism, given that *C. elegans* has acquired the capability to produce a signaling molecule, via HGT from bacteria, that mimics the bacteria-dependent becyp#1[98]. Lastly, the biogenesis of becyp#1 highlights the significance of microbiota-dependent small molecule signals that arise from shared biochemical networks involving both metazoan and bacterial metabolism[99,100], a challenge that we here show can be addressed by whole organism comparative metabolomics.

## Methods
### Nematode strains

Unless otherwise indicated, worms were maintained on Nematode Growth Medium (NGM) 6 cm diameter petri dish plates seeded with *E. coli* OP50 obtained from the *Caenorhabditis* Genetics Center (CGC), except for the *acdh-11(n5878)* mutant, which was maintained on cyclopropane-deficient JW1653-1. For imaging experiments, worms

were grown on 3.5 cm diameter petri dish plates seeded with *E. coli* OP50, HB101, BW25113, JW1653-1, or HT115 for RNAi, as indicated. The following Nematode strains were used for comparative metabolomics: *C. elegans* Bristol N2 ("wildtype"), *C. elegans* FCS7 *hacl-1(tm6725)* II, *C. elegans* FCS40 *fcmt-1(gk155709)* II, *C. elegans* FCS20 *fcmt-1(tm2382)* II, *C. elegans* DMS303 *nIs590[P_{fat-7}::fat-7::GFP]* V ("wildtype"), *C. elegans* DMS441 *acdh-11(n5878)* III; *nIs590* V, *C. elegans* FCS66 *hacl-1(tm6725)* II; *nIs590* V, *C. elegans* BX115 *lin-15B&lin-15A(n765)* X; *waEx16[P_{fat-6}::fat-6::GFP + lin15(+)]*, and *C. Briggsae* AF16 ("wildtype"). *hacl-1* mutants were backcrossed as previously described[47], *fcmt-1* mutants were backcrossed with Bristol N2 for a total of six generations, and other strains were analyzed as received from the CGC. The genotypes were confirmed by PCR and Sanger sequencing (Cornell University Institute of Biotechnology). Primers (Integrated DNA Technologies) used for genotyping are listed in Supplementary Table 3.

Of note, several unrelated metabolites were significantly enriched only in the FCS20 *fcmt-1(tm2382)* strain, but not FCS40 *fcmt-1(gk155709)* (Supplementary Fig. 10). The *tm2382* deletion affects a genomic interval encoding genes on both strands of DNA and deletes intronic sequence of *VF13D12L.3*, which encodes an enzyme annotated as a malate/lactate oxidoreductase. To avoid potentially confounding effects resulting from presumed disruption of *VF13D12L.3*, subsequent experiments were conducted using the *fcmt-1(gk155709)* mutant.

## Metabolite nomenclature

All newly detected metabolites for which a structure could be proposed were named using Small Molecule Identifiers (SMIDs), a search-compatible nomenclature for metabolites identified from *C. elegans* and other nematodes. The SMID database (www.smid-db.org) is an electronic resource maintained in collaboration with WormBase (www.wormbase.org); a complete list of SMIDs can be found at www.smid-db.org/browse.

## *C. elegans* liquid cultures

Alkaline bleach treatment of mixed-stage animals yielded a sterile egg suspension, which was rocked overnight in 3–5 mL M9 solution to yield synchronized, starved L1 larvae[101]. For the analysis of staged adults, cultures ranging from 25,000-75,000 synchronized L1 larvae obtained from alkaline bleach treatment were added to 50- or 125 mL Erlenmeyer flasks containing S-complete medium such that the density of worms was maintained at approximately 3000 nematodes / mL, and kanamycin added at 35 μg/mL to prevent contamination. Worms were fed with 50x concentrated *E. coli* and incubated at 20 °C with shaking at 180 RPM for 64–70 h, unless otherwise indicated. Control samples to account for bacterial matrix were prepared with the same amount of *E. coli* under identical conditions. For the analysis of mixed-stage cultures, a well-populated 6 cm NGM plate was washed with M9 solution, and the recovered animals seeded to liquid cultures and incubated with shaking for 2–7 days, as described above. Cultures were fed with additional bacteria as needed, determined by cloudiness of cultures. Liquid cultures were transferred to 15- or 50 mL conical tubes and centrifuged (200 × *g*, 22 °C, 30 s), and the supernatant (*exo*-metabolome) was transferred to a fresh conical tube and snap frozen. Remaining worm pellet (*endo*-metabolome) was transferred to a 15 mL conical tube and washed with 10 mL M9 solution, centrifuged (200 × *g*, 22 °C, 30 s), supernatant removed, and worm pellet washed twice more with M9 solution before snap freezing in liquid nitrogen.

## Stable isotope labeling experiments

Alkaline bleach treatment of a mixed-stage *hacl-1(tm6725)* culture yielded a sterile egg suspension that was evenly divided into three 50 mL Erlenmeyer flasks containing 10 mL M9 solution. Worm suspensions were supplemented with water (vehicle control), methionine (Sigma M-9625), or isotopically labeled $D_3$-methyl-methionine (Cambridge Isotope Laboratories DLM-431-1) at a final concentration of 5 mM. Cultures were incubated at 20 °C with shaking at 180 RPM for approximately 24 h. *Exo*- and *endo*-metabolome samples were harvested separately, as described above. Two independent labeling experiments in *hacl-1(tm6725)* were performed, one in larvae as described and one in mixed-stage cultures, described below.

Mixed-stage cultures of N2 and *hacl-1(tm6725)* were collected, washed in M9 solution, and evenly divided to five or seven 50 mL Erlenmeyer flasks containing 10 mL S-Complete medium and kanamycin at 35 μg/mL. Worms were fed with 50x concentrated *E. coli* and supplemented with ethanol (vehicle control), *cis*-vaccenic acid (Cayman Chemical 20023), $D_{13}$-*cis*-vaccenic acid (Cayman Chemical 27716), *trans*-vaccenic acid (Cayman Chemical 15301), or $D_{13}$-*trans*-vaccenic acid (Cayman Chemical 27717) at a final concentration of 100 μM (first replicate) or 75 μM (second replicate); *hacl-1(tm6725)* was additionally supplemented with methionine or $D_3$-methyl-methionine at a final concentration of 5 mM. Cultures were incubated at 20 °C with shaking at 180 RPM for 3-4 days. *Exo*- and *endo*-metabolome samples were harvested separately, as described above. One labeling experiment was performed in N2 (WT), and two independent labeling experiments were performed in *hacl-1(tm6725)* mutant.

## Evolutionary analysis of ACDH-11 and FCMT-1 homologs

PSI-BLAST searches on the NCBI-nr database using the *C. elegans* ACDH-11 sequence (NP_001033378.1) as query were run on July 23, 2023. The first search was not restricted to any taxonomy and returned hits predominantly from clade V (Rhabditina), from which the satellite model organisms *Caenorhabditis briggsae* and *Pristionchus pacificus* were selected for alignment on the basis of high-quality genomic data, as well as two representative hits from the Strongylida suborder (*Ancylostoma ceylanicum* and *Haemonchus contortus*). An additional PSI-BLAST was conducted using the *C. elegans* ACDH-11 sequence as query and excluding the subclass Rhabditina (taxid:2301116), which returned hits belonging to clade IV (Tylenchina) nematodes from the *Strongyloididae* and *Steinernematidae* families, which are represented by ACDH-11 sequences from *Strongyloides ratti* and *Steinernema carpocapsae*, respectively[102]. To compare against other ACDH enzymes in *C. elegans*, a second PSI-blast search was performed using the *C. elegans* ACDH-11 sequence as query restricted to *C. elegans* (taxid: 6239), and all hits above the minimum threshold were downloaded. A MUSCLE alignment of ACDH-11 and the 19 selected sequences was performed in MEGA 11 software using default settings[103]. The evolutionary history was inferred by using the Maximum Likelihood method and LG + G model with 200 bootstrap iterations in MEGA 11 software[104]. The tree with the highest log likelihood (-19853.00) is shown. Initial tree(s) for the heuristic search were obtained automatically by applying Neighbor-Join and BioNJ algorithms to a matrix of pairwise distances estimated using the JTT model, and then selecting the topology with superior log likelihood value. A discrete Gamma distribution was used to model evolutionary rate differences among sites (5 categories, +*G* parameter = 1.8549). The tree is drawn to scale, with branch lengths measured in the number of substitutions per site.

Blastp searches on the NCBI-nr database using the *C. elegans* FCMT-1 sequence (NP_001380126.2) as query were run on August 7, 2022. The first search was not restricted to any taxonomy. Sequences of the top 50 blast hits from this search were downloaded and included 37 nematode and 13 bacterial sequences. To compare homologs in other animals, we performed a second blastp search restricted to Animalia (taxid: 33208) but excluding Nematoda (taxid: 6231). The top 10 sequences from this search were also downloaded. A MUSCLE alignment of FCMT-1 and the 60 sequences from both searches was performed in MEGA 11 software using default settings[103]. The evolutionary history was inferred by using the Maximum Likelihood method and JTT matrix-based model with 200 bootstrap iterations in MEGA 11 software[105]. The tree with the highest log likelihood (-46241.85) is shown. Initial tree(s) generation and branch lengths as described above.

## FAT-7::GFP microscopy

Commercially available and synthetic compounds were prepared as stock solutions at 10 mg/mL in ethanol or DMSO, as indicated. Stock solutions were diluted into 40 μL of a 50:50 mixture of solvent: water or 40 μL of solvent only, as indicated, then applied topically to NGM 3.5 cm diameter petri dish plates (4 mL NGM agar volume) seeded with *E. coli* and allowed to dry with the lid off for 15 min in a laminar flow cabinet. The lids were replaced, and the plates were shifted to an incubator at the experimental temperature and stored for 1–2 h before plating synchronized, starved L1 larvae obtained from alkaline bleach treatment of a 6 cm NGM maintenance plate. Alternatively, young adult animals were placed on treatment plates and allowed to lay eggs for 4–6 h, the adults removed, and the resulting progeny analyzed.

Gravid adult animals were removed from treatment plates by washing with M9 buffer and collecting in 1.5 mL Eppendorf tubes. Samples were centrifuged (100 × *g*, 22 °C, 30 s) using an Eppendorf benchtop centrifuge. The supernatant was aspirated, and animals were washed with 1 mL of M9 buffer, centrifuged, and the supernatant aspirated to leave approximately 0.1 mL remaining. An equal volume (-0.1 mL) of 20 mM sodium azide in M9 was added to the tubes to anaesthetize live animals. A -5–10 μL aliquot of the paralyzed worms was then dropped onto an unseeded NGM plate, and the liquid was allowed to fully absorb into the agar. Worms were imaged directly on NGM plates using a Leica M205FA stereomicroscope outfitted with a DFC7000T camera and controlled by Leica Application Suite X software (v3.6.0.20104 Leica Microsystems). Transmitted light brightfield images were acquired with no filter; incident light was generated by a Lumencor Sola Light Engine (SM5-LCR-VA Lumencor, Inc.) and GFP fluorescence was acquired with an ET GFP bandpass filter, excitation 450–490 nm, emission 500–550 nm (Leica Microsystems part no. 10447408). Typical microscope settings were as follows: magnification 75x; transmitted light: exposure 20 ms, gain 1, intensity 55%, aperture 60%, auto balance off; incident light: exposure 400 ms, gain 3, intensity 100%, aperture N/A. The magnification and exposure settings were consistent within each experiment and any quantification was performed relative to a control performed in parallel. Quantification was performed using ImageJ software (v1.54c). In brief, individual animals were outlined using the *freehand selection* tool in the brightfield micrographs, and then these outlines were transferred to the GFP micrographs using the *restore selection* command. The intensity of the outlined area was analyzed using the *measure* command. Background intensity was calculated by averaging the minimum intensity of all outlined areas in a given experiment; this value was subtracted from the mean intensity for each individual measurement. Intensities were normalized to the average of the untreated control in each independent experiment.

## Gene expression analysis

Stock solutions were diluted as described above and treatment plates were prepared by applying 80 μL of a 50:50 mixture of DMSO: water (vehicle) or the same mixture containing (*R*)-bemeth#1 such that the final concentration on the plates was 0.05 mM. Solutions were applied topically to NGM 6 cm petri dish plates (8 mL NGM agar volume) seeded with *E. coli* JW1653-1 and allowed to dry and equilibrate, as described above. Alkaline bleach treatment of mixed-stage animals yielded a sterile egg suspension. Following counting and concentration, 2000 eggs were applied directly to each of four plates per condition per experiment. Eggs were incubated on plates at 20 °C for 70–74 h, at which point gravid adult animals were removed by washing with M9 buffer and collecting in 1.5 mL Eppendorf tubes. Samples were centrifuged (100 × *g*, 22 °C, 30 s) using an Eppendorf benchtop centrifuge and washed twice, as described above, before snap freezing in liquid nitrogen and storage at −80 °C. Total RNA was extracted and purified using Trizol reagent and RNA Clean & Concentrator-5 kit (Zymo Research cat. no. R1013) according to the manufacturer's

protocol. Following quantification, 2 μg of RNA per sample were used to synthesize cDNA using the SuperScript III First-Strand kit (Invitrogen cat. no. 12574026). RT-PCR was performed using SYBR green dye (Thermo Fisher Scientific cat. no. 4367659) and a Bio-Rad C1000TM Thermal Cycler. Relative gene expression levels were calculated using the ΔΔCt method with *act-1* as the reference gene[106]. Four independent biological experiments were performed and analyzed as three technical replicates per gene per experiment in 10 μL reaction volumes. Technical replicates were averaged prior to calculating ΔΔCt values, which were normalized against the average of all mock (control) samples. Primers (Integrated DNA Technologies) used for RT-PCR are listed in Supplementary Table 4.

## Cyclopropyl fatty acid supplement

Mixed-stage cultures of $P_{fat-7}::fat-7::GFP$ (WT) and *acdh-11(n5878); $P_{fat-7}::fat-7::GFP$* reared on JW1653-1 (*Δcfa*) *E. coli* were collected, washed in M9 solution, and evenly divided to four 50 mL Erlenmeyer flasks containing 10 mL S-Complete medium and kanamycin at 35 μg/mL. Worms were fed with 50x concentrated *E. coli* BW25113 (WT) or (*Δcfa*) *E. coli* and supplemented with ethanol (vehicle control), lactobacillic acid (Cayman Chemical 10012556), or dihydrosterculic acid (Cayman Chemical 24824) at a final concentration of 20 μM. Cultures were incubated at 20 °C with shaking at 180 RPM for 3–4 days. *Exo-* and *endo-*metabolome samples were harvested separately, as described above.

## Sample preparation for HPLC-HRMS

Animal bodies (*endo*-metabolome) and conditioned medium (*exo*-metabolome) were frozen and processed separately. For preparation of *endo*-metabolome extracts, samples were lyophilized for 18–24 h using a VirTis BenchTop 4 K Freeze Dryer. After the addition of 1 mL methanol directly to the conical tube in which animals were frozen, samples were sonicated for 5 min (2 s on/off pulse cycle at 90 A) using a Qsonica Q700 Ultrasonic Processor with a water bath cup horn adaptor (Qsonica 431C2). Following sonication, an additional 4-9 mL of methanol was added, depending on sample size, and the extract rocked overnight at room temperature. The conical tubes were centrifuged (3000 × *g*, 22 °C, 5 min) and the resulting clarified supernatant transferred to a clean 8- or 20-mL glass vial which was concentrated to dryness in an SC250EXP Speedvac Concentrator coupled to an RVT5105 Refrigerated Vapor Trap (Thermo Scientific). The resulting powder was suspended in 100–250 μL of methanol, depending on sample size, followed by vigorous vortex and brief sonication. This solution was transferred to a clean microfuge tube and subjected to centrifugation (20,000 × *g*, 22 °C, 5 min) in an Eppendorf 5417 R centrifuge to remove precipitate. The resulting supernatant was transferred to an HPLC vial and analyzed by HPLC−HRMS.

For preparation of *exo*-metabolome extracts, samples were lyophilized -48 h using a VirTis BenchTop 4 K Freeze Dryer. Dried material was extracted in 5–15 mL methanol, depending on sample size, and rocked overnight at room temperature. The conical tubes were centrifuged (3000 × *g*, 22 °C, 5 min) and the resulting clarified supernatant transferred to clean 8- or 20-mL glass vials which were concentrated *in vacuo* and suspended in methanol as described for *endo*-metabolome samples.

## HPLC-HRMS analysis

Reversed-phase chromatography was performed using a Vanquish HPLC system controlled by Chromeleon Software (ThermoFisher Scientific) and coupled to an Orbitrap Q-Exactive HF mass spectrometer controlled by Xcalibur software (ThermoFisher Scientific), or by a Dionex Ultimate 3000 HPLC system coupled to an Oribtrap Q-Exactive mass spectrometer controlled by the same software. Extracts prepared as described above were separated on a Thermo Scientific Hypersil Gold column (150 mm × 2.1 mm, particle size 1.9 μm, part no. 25002-

152130) maintained at 40 °C with a flow rate of 0.5 mL/min. Solvent A: 0.1% formic acid (Fisher Chemical Optima LC/MS grade; A11750) in water (Fisher Chemical Optima LC/MS grade; W6-4); solvent B: 0.1% formic acid in acetonitrile (Fisher Chemical Optima LC/MS grade; A955-4). A/B gradient started at 1% B for 3 min after injection and increased linearly to 98% B at 20 min, followed by 5 min at 98% B, then back to 1% B over 0.1 min and finally held at 1% B for an additional 2.9 min.

Reversed-phase post-column ion-pairing chromatography was performed using the same system as described; extracts were separated on a Thermo Scientific Hypersil Gold column (150 mm × 2.1 mm, particle size 1.9 μm, part no. 25002-152130) or on a Kinetex Evo C18 (150 mm × 2.1 mm, particle size 1.7 μm, part no. 00F-4726-AN) maintained at 40 °C with a flow rate of 0.5 mL/min. Solvent A: 0.1% ammonium acetate in water; solvent B: acetonitrile. A/B gradient started at 5% B for 3 min after injection and increased linearly to 98% B at 20 min, followed by 5 min at 98% B, then back to 5% B over 0.1 min and finally held at 5% B for an additional 2.9 min. A second pump (Dionex 3000) controlling a solution of 800 mM ammonia in methanol was run at a constant flow rate of 0.015 mL/min for the duration of the method and mixed via micro-splitter valve (Idex #P-460S) with the eluate line from the column.

Mass spectrometer parameters: spray voltage, −3.0 kV/+3.5 kV; capillary temperature 380 °C; probe heater temperature 400 °C; sheath, auxiliary, and sweep gas, 60, 20, and 2 AU, respectively; S-Lens RF level, 50; resolution, 60,000 or 120,000 at $m/z$ 200; AGC target, 3E6. Each sample was analyzed in negative (ESI−) and positive (ESI+) electrospray ionization modes with $m/z$ range 117–1000. Parameters for MS/MS (dd-MS2): MS1 resolution, 60,000; AGC Target, 1E6. MS2 resolution, 30,000; AGC Target, 2E5. Maximum injection time, 60 msec; Isolation window, 1.0 $m/z$; stepped normalized collision energy (NCE) 10, 30; dynamic exclusion, 1.5 sec; top 8 masses selected for MS/MS per scan.

HPLC-HRMS RAW data were converted to mzXML file format using MSConvert (v3.0, ProteoWizard) and were analyzed using Metaboseek software (v0.9.9.0) with the following settings: 5 ppm, 3_20 peakwidth, 3 snthresh, 3_100 prefilter, FALSE fitgauss, 1 integrate, TRUE firstBaselineCheck, 0 noise, wMean mzCenterFun, −0.005 mzdiff. Default settings for XCMS feature grouping: 0.2 minfrac, 2 bw, 0.002 mzwid, 500 max, 1 minsamp, FALSE usegroups. Metaboseek peak filling used the following settings: 3 ppm_m, 3 rtw, TRUE rtrange, FALSE areaMode. Data were normalized to the abundance of ascr#3 in negative ionization mode or normalized to the abundance of ascr#2 in positive ionization mode. Quantification was performed with Metaboseek software or via integration using Xcalibur QualBrowser v4.1.31.9 (Thermo Fisher Scientific) using a 3-ppm window around the $m/z$ of interest.

For volcano plot in Fig. 1D, the *exo*-metabolome in negative ionization mode is depicted as a representative dataset. The list of features identified by peak picking in Metaboseek (including degenerate features such as adducts and isotopes) was culled by retention time (180–1080 s) and further filtered by grouped analysis as described[47]. In brief, all *C. elegans* samples were grouped and compared against a blank solvent injection. Blank subtraction was performed by removing any feature less than five-fold more abundant in *C. elegans* samples relative to blank. Features were further culled by a mean intensity threshold of 10,000 AU for the *C. elegans* group. The resulting feature list of 30,191 features was regrouped according to genotype and analyzed by unpaired, two-sided Welch's t-test; because ascr#3 was used as a normalization factor, ascr#3 isotopes and adducts were manually removed. A representative experiment comparing $P_{fat-7}::fat-7::GFP$ WT reporter strain and the *acdh-11(n5878)*;$P_{fat-7}::fat-7::GFP$ mutant at different temperatures was analyzed, comprising two cultures reared at 20 °C and one culture each at 15 °C and 25 °C, which were modeled as four independent experiments.

For volcano plots in Fig. 4E and Supplementary Fig. 11, the *exo*-metabolome in negative ionization mode is depicted. Data represents three independent experiments each with two technical replicates grown and extracted independently. For purpose of comparison, data were modeled as six independent experiments (or five for *fcmt-1(gk155709)*) and adjusted for significance by the Benjamini-Hochberg method using a false discovery rate of 15%, as indicated[107]. List of features culled as above except retention time window adjusted (60–1080 s). The resulting feature list of 40,436 features was analyzed by unpaired, two-sided Welch's t-test; ascr#3 isotopes and adducts were manually removed, as above. Statistical analysis for metabolomics was performed with Metaboseek software (v0.9.9.0), Microsoft Excel (v2302 Build 16.0.16130.20332), and with GraphPad Prism (v9.5.0.730).

### Isolation and NMR spectroscopy of becyp#2
The *exo*-metabolomes of several medium-scale *C. elegans* cultures were lyophilized and extracted with methanol, as described. Dried methanol extract was loaded onto Celite and fractionated using medium pressure reverse phase chromatography (15 g C18 Combiflash RediSep column, Teledyne Isco 69-2203-334). Solvent A: 0.1% acetic acid in water; solvent B: acetonitrile. Column was primed with 1% B; separation was achieved by 5% B for 2 column volumes (CV), which was increased linearly to 50% B over 15 CV, then to 100% B over 3 CV and held at 100% B for 5 CV, before returning to 80% B for 3 CV. Fractions were assayed for compounds of interest by HPLC-MS, the relevant fractions were combined and dried *in vacuo*. Following suspension in water: methanol (1:2), the pooled fractions were further separated by semi-preparative HPLC on a Thermo Hypersil Gold C18 column (250 mm × 10 mm, particle size 5 μM; 25005-259070) using a Vanquish UPLC system controlled by Chromeleon Software (ThermoFisher Scientific) and coupled to a Dionex UltiMate 3000 Automated fraction collector and to an Orbitrap Q-Exactive High Field mass spectrometer using a 9:1 split. Fractions containing becyp#2 were combined and analyzed by 2D NMR spectroscopy (CD$_3$OD, Bruker Avance III HD, 800 MHz). For NMR spectroscopic data, see Supplementary Table 5.

### Statistics and reproducibility
No statistical method was used to predetermine sample size. No data were excluded from the analyses. The experiments were not randomized. The investigators were not blinded to allocation during experiments and outcome assessment.

### Chemical syntheses
See "Supplementary Information" for synthetic schemes, synthesis procedures, and NMR spectroscopic data.

### Reporting summary
Further information on research design is available in the Nature Portfolio Reporting Summary linked to this article.

## Data availability
The HPLC-HRMS data generated during this study have been deposited in the MassIVE database under accession code MSV000092700 [https://doi.org/10.25345/C50R9MF18]. Detailed information about newly described C. elegans compounds can be accessed via the Small Molecule Identifier Database (SMID-DB, https://www.smid-db.org/). Protein sequence data was downloaded from the National Center for Biotechnology Information (NCBI) non-redundant (nr) database (https://www.ncbi.nlm.nih.gov/refseq/about/nonredundantproteins/). Source data are provided with this paper.

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

## Acknowledgements

This research was supported in part by the National Institutes of Health (R35GM131877 to F.C.S.) and the Howard Hughes Medical Institute (Faculty Scholar grant to F.C.S.). MJH received a Research Fellowship from the Deutsche Forschungsgemeinschaft (D.F.G.), Project Number 386228702. Some strains used in this work were provided by the CGC, which is funded by the NIH Office of Research Infrastructure Programs (P40 OD010440). hacl-1(tm6725) and fcmt-1(tm2382) were obtained from the National Bioresource Project, Tokyo, Japan. We thank Gary Horvath for technical support. We thank Camila Castellanos for assistance with sample processing and Erich Schwarz for helpful discussions and critical feedback. This work was also supported by the National Science Foundation (Award 2042101 to F.C.S.).

## Author contributions

F.C.S. and S.S.L. supervised the study. B.W.F., R.N.B., M.J.H., A.B.A., D.F.P., A.C., A.T., A.F.S. and C.J.J.W. performed chemical and biological experiments. R.N.B., B.J.C. and Y.K.Z. performed syntheses. B.W.F. and F.C.S. wrote the paper with input from all authors.

## Competing interests

F.C.S. is a co-founder of Holoclara and Ascribe Bioscience, a member of the Board of Directors of Ascribe Bioscience, and a member of the Scientific Advisory Board of Hexagon Bio. The remaining authors declare no competing interests.
