## [Peer Review File · Nature Communications]

REVIEWER COMMENTS

Reviewer #1 (Remarks to the Author):

In this study, Fox et al. discover parallel pathways in metazoa and bacteria which converge to regulate fatty acid desaturation. Specifically, they demonstrate that an endogenous and microbiota-dependent small molecule signals promote lipid desaturation via the nuclear receptor NHR-49/PPAR α in *C. elegans*. The conclusion they make is that there are evolutionarily related biosynthetic pathways in metazoan host and associated microbiota which can converge to both regulate fat desaturation and degrade through independent pathways of oxidation. This is mediated by 2 similar cyclopropane fatty acids produced by host and bacteria.

These results have the potential to be of significance to the lipid signaling and metabolism field. I found this paper technically sound and their main conclusions supported by the data provided. The study is logically presented, and the chemistry aspect is strong, however the biological relevance and applications of the findings could be further expanded upon to be more applicable to a broad readership and communicate the findings in the context of mammalian biology. To this end, I have added some comments I believe are required to be addressed before publication.

Major Comments

1. There is a large missed opportunity here to connect these findings to human biology. SCD-1 is also found in humans, as are cyclopropane fatty acids. Given the large impact of unsaturated fatty acids on human metabolism, the authors fail to put their discovery into context to humans. This can be possible given the large databases of dietary, microbiome and human associated metabolites publicly available, where the *E. coli* cyclopropanes described here could be found. Given the major conclusion of evolutionary conservation, and the conservation of the genes listed in *C. elegans* and humans, it stands to reason that the authors should do a better job of putting their data in context of human biology through the introduction, discussion and also data collection of the enzymes showing homology to humans.

2. Can you find bemeth#1, becyp#1 and/or related cyclopropane fatty acids you structurally characterized here in the gut microbiome of humans? How much and is it increased in disease? There are many open-source metabolomic datasets this could be assessed. Also you need to check metabolomic FoodDB to get a more accurate read which foods it is in

3. Connected to this point, there needs to be data added as a supplement to Figure 4 and 5 of how mammalian conservation of these pathways look from an enzymatic perspective and metabolite network

4. In the *E. coli* metabolomics experiment, it is logical that the becyp#1 source is from *E. coli* cyclolipids given the experiment however there is an argument to be made the loop is not closed here. Was their data on *E. coli* alone in culture WT vs delta-CFA (cyclopropane lipid free) without the host colonization? Can you confirm the lack of cyclopropane lipids without the host interference and which cyclopropane lipids to *E. coli* produce? Without these, it is not 100% conclusive that its not a host factor interacting with the mutant in a different way than the WT bacteria.

5. Can you demonstrate that becyp#1 and bemeth#1 regulates fat-7 expression in an NHR-49/PPAR-dependent manner in mammalian cells using qPCR or RNA-sequencing?

6. Can you remark on how the presence or absence of cyclopropane lipids from *E. coli* in the mutant experiment (*C. elegans* +/- WT or *cfa* mutant) alters the lipidome of *C. elegans* itself other than Saturated to Unsaturated pathways from Fat7 but PUFAs and ratios of other lipids? Do *E. coli* lipids integrate into *C. elegans* cells in WT vs *cfa* mutant? A full picture of the complexity of these microbe-host lipid interactions is missing or buried in the metabolomic raw data from the experiment.

Minor Comments

1. In the discussion, line 301, it is listed that the link to human biology in the diet is cyclopropane lipids are more often found in animal products. This links to a paper showing silage grain fed cattle have cyclopropane in their cheese compared to cattle allowed to forage on grass, and used as a biomarker for this. To me this misses a major point. One, is that you can find cyclopropane lipids widely within humans from their own microbiome and secondly they are also found in plants. The authors should make it clear that these cyclopropane lipids that promote lipid desaturation in *C. elegans* are not becoming more abundant in the diet because of animal products per say but the grains they are fed and they are also common in plants and human endogenous microbiome.

2. In general, more should be done in the discussion to put the findings into context for mammalian biology; what are the limitations in applicability to other organisms aside from *C. elegans* of these discovered interactions

3. Figure 1D the volcano plot is poorly labeled, which metabolites there are most up and downregulated (label this)

4. Can you comment on the localization of Fat7-GFP to the polar ends of *C. elegans* after *becyp#1* addition? Is this biologically significant? The expression is not homogenous as it is predicted to be in figure 1B

5. The staining in Figure 2 in Fat7 is different than Figure 1 how do you explain this within the model of *becyp#1* addition driving Fat7 is it tissue specific? There is also no mention of the localization of the Fat7 signal in Figure 3

6. In Figure 4A which enzyme converts oleic acid to PUFAs? Are these enzymes missing because they are unknown?

Reviewer #2 (Remarks to the Author):

This is a fascinating study that breaks new ground in the regulation of fatty acid desaturation. The ratio of saturated to unsaturated fats in membranes of *C. elegans*, which is important to maintain membrane fluidity in changing environments, has been shown to involve several nuclear hormone receptors regulating the Δ -9 fatty acid desaturase enzymes. This study used exquisite stable isotope and mass spec techniques to discover several ligands that activate the nuclear hormone receptor NHR-49 in response to nutritional cues. Interestingly, one of the ligands is a bacterial fatty acid metabolite and the other is an endogenously synthesized metabolite. During the course of this work, the authors discovered that one *C. elegans* acyl-coA dehydrogenase, *acdH-11*, has specialized activity that is required for the breakdown of cyclopropyl fatty acids, fatty acids that are particularly large dietary components of lab-grown *C. elegans* due to their abundance in the common *C. elegans* food *E. coli* OP50. The other groundbreaking finding is discovery of a unique nematode methyl transferase that likely evolved via horizontal gene transfer between bacteria and nematodes. This methyl transferase, FCMT-1, is required to synthesize the endogenous ligand that activates NHR-49. The experiments are thorough, carefully done and beautifully presented, and the interpretations are

supported by the experimental data. I learned a lot about specialized fatty acid breakdown pathways reading this paper, it truly represents an important addition to lipid metabolism literature.

We would like to thank the editor and the reviewers for their evaluation of our manuscript. Please see our point-by-point responses below.

Reviewers' comments:

Reviewer #1 (Remarks to the Author):

*In this study, Fox et al. discover parallel pathways in metazoa and bacteria which converge to regulate fatty acid desaturation. Specifically, they demonstrate that an endogenous and microbiota-dependent small molecule signals promote lipid desaturation via the nuclear receptor NHR-49/PPAR α in *C. elegans*. The conclusion they make is that there are evolutionarily related biosynthetic pathways in metazoan host and associated microbiota which can converge to both regulate fat desaturation and degrade through independent pathways of oxidation. This is mediated by 2 similar cyclopropane fatty acids produced by host and bacteria.*

These results have the potential to be of significance to the lipid signaling and metabolism field. I found this paper technically sound and their main conclusions supported by the data provided. The study is logically presented, and the chemistry aspect is strong, however the biological relevance and applications of the findings could be further expanded upon to be more applicable to a broad readership and communicate the findings in the context of mammalian biology. To this end, I have added some comments I believe are required to be addressed before publication.

Response: We appreciate this reviewer's thoughtful comments.

Major Comments

*1. There is a large missed opportunity here to connect these findings to human biology. SCD-1 is also found in humans, as are cyclopropane fatty acids. Given the large impact of unsaturated fatty acids on human metabolism, the authors fail to put their discovery into context to humans. This can be possible given the large databases of dietary, microbiome and human associated metabolites publicly available, where the *E. coli* cyclopropanes described here could be found.*

Response: We thank the reviewer for this suggestion. We have searched the Human Metabolome Database (HMDB, PMID: 34986597), the Global Natural Product Social Molecular Networking (GNPS) spectral libraries (PMID: 27504778), the Curated Gut Microbiome-Metabolome Data Resource (PMID: 36243731), the Human Microbial Metabolome Database (MiMeDB, PMID: 36215042), the Fecal Metabolome Database (PMID: 30032758), and FooDB, as suggested below. While we did not find evidence for the specific cyclopropyl metabolites we identify in this study (see below for details), we have revised the discussion to forge a stronger connection between bacterial cyclopropane fatty acids and their potential impact on host biology, including in humans (see Lines 318-339).

*Given the major conclusion of evolutionary conservation, and the conservation of the genes listed in *C. elegans* and humans, it stands to reason that the authors should do a better job of putting their data in context of human biology through the introduction, discussion and also data collection of the enzymes showing homology to humans.*

Response: The major evolutionary relationship explored in the manuscript is between bacterial cyclopropane synthase and a methyltransferase found in nematodes, but not in humans (Figure 4C). There are no homologous methyltransferases in humans. Furthermore, we have now clarified in the text that the *acdH-11* gene is likely not conserved in humans (see Line 341). This

lack of enzyme conservation begs an important question – how are cyclopropane lipids metabolized in animals that do not have a specialized enzyme like ACDH-11?

2. *Can you find bemeth#1, becyp#1 and/or related cyclopropane fatty acids you structurally characterized here in the gut microbiome of humans? How much and is it increased in disease? There are many open-source metabolomic datasets this could be assessed. Also you need to check metabolomic FoodDB to get a more accurate read which foods it is in*

Response: The fatty acids bemeth#1 and becyp#1 are not abundant and do not ionize well under standard conditions, which is perhaps why they have not been previously identified.

We found one reported association between cyclopropyl fatty acids and disease states in humans (PMID 21687748), and we agree that it would be important to explore the production of cyclopropane lipids in the gut by human-associated gut microbiota. That there have not been any follow-up studies on cyclopropyl fatty acids as disease biomarkers seems like an important opportunity, which we have now included in the discussion (see Lines 336-339).

Searching by molecular weight in FoodDB revealed some interesting fatty acids of similar length, including naturally occurring undecylenic acid (FDB011844). There are entries for (Z)-3-Methyl-3-decenoic acid (FDB003064) and (Z)-3-Methyl-4-decenoic acid (FDB003065), the latter of which is the geometric isomer of bemeth#1; however, there is no information about its occurrence in mammalian biofluids. Additional searching yielded cascarillic acid, the *trans* isomer of becyp#1 obtained from cascarilla essential oil (DOI: 10.1016/j.tetlet.2004.09.141), two medium-chain cyclopropane fatty acids isolated from a marine bacterium (PMID: 30297608), and an unusual cohort of methyl-cyclopropyl fatty acids from a deep-sea fungus (PMID: 35877703). We did not think it would make sense to add these references; as mentioned above, our study provides strong motivation to explore the fate of cyclopropyl and related methyl branched fatty acids in humans.

3. *Connected to this point, there needs to be data added as a supplement to Figure 4 and 5 of how mammalian conservation of these pathways look from an enzymatic perspective and metabolite network*

Response: We have color-coded homologous enzymes and included in the legend of Figure 3A and in the main text that α -oxidation is highly conserved between *C. elegans* and humans. As pointed out above, there are no similar methyltransferases in humans. α -oxidation in humans is best characterized for its role in the degradation of phytanic acid, derived from phytol from the breakdown of chlorophyll (see reference 72, Lines 307-308).

4. *In the E. coli metabolomics experiment, it is logical that the becyp#1 source is from E. coli cyclolipids given the experiment however there is an argument to be made the loop is not closed here. Was their data on E. coli alone in culture WT vs delta-CFA (cyclopropane lipid free) without the host colonization? Can you confirm the lack of cyclopropane lipids without the host interference and which cyclopropane lipids to E. coli produce? Without these, it is not 100% conclusive that its not a host factor interacting with the mutant in a different way than the WT bacteria.*

Response: In earlier work, we performed 2D-NMR (dqfCOSY) analyses of extracts of Δcfa *E. coli* and of *C. elegans* fed Δcfa *E. coli*, neither of which exhibited ^1H - ^1H cross peaks corresponding to the cyclopropane ring. These samples were compared with WT *E. coli* and with *C. elegans* fed WT *E. coli*, both of which display strong ^1H - ^1H cross peaks corresponding to the cyclopropyl protons (Please see PMID 35145075, Supplementary Figure 12 for the worm

extracts). *E. coli* predominantly produce two cyclopropane-containing acyl groups in the context of membrane lipids (the majority being phosphatidylethanolamine), *cis*-11,12-methyleneoctadecanoic acid (known as lactobacillic acid) and *cis*-9,10-methylenehexadecanoic acid (PMID: 9409147).

5. *Can you demonstrate that becyp#1 and bemeth#1 regulates fat-7 expression in an NHR-49/PPAR-dependent manner in mammalian cells using qPCR or RNA-sequencing?*

Response: This is a great idea, and we intend to follow up on this suggestion, but we respectfully argue that this is a highly ambitious experiment that is outside of the scope of the current manuscript. For example, in order to translate this experiment into mammalian cells, one may need to transfect additional components of the transcriptional machinery. To our knowledge, full reconstitution of *C. elegans* NHR activation in mammalian cell culture has been achieved only in the case of *daf-12*, which was a fortuitous case because the mammalian SRC-1 co-activator was functional (PMID 16529801, 22170062, 24411940)

6. *Can you remark on how the presence or absence of cyclopropane lipids from E. coli in the mutant experiment (C. elegans +/- WT or cfa mutant) alters the lipidome of C. elegans itself other than Saturated to Unsaturated pathways from Fat7 but PUFAs and ratios of other lipids?*

Response: We thank the reviewer for this excellent question and have performed additional experiments to study the impact of dietary cyclopropyl lipids on *C. elegans* lipid metabolism.

We added data showing that the metabolomes of *C. elegans* fed WT *E. coli* or Δcfa *E. coli* exhibit dramatic differences. These included several families of metabolites derived from incorporation of bacterial cyclopropane fatty acids into host-dependent lipids, for example series of lysophosphatidylcholines (PMID: 22922101) and *N*-acyl glycolyglycerophosphoethanolamines (PMID: 35145075), which were abundant in *C. elegans* fed WT *E. coli* but abolished in animals fed Δcfa *E. coli*. Comparative analyses of free fatty acid profiles further revealed large increases in monounsaturated vaccenic acid and palmitoleic acid in animals reared on Δcfa *E. coli*, regardless of *C. elegans* genotype and consistent with a previous report (PMID: 24475206). Vaccenic and palmitoleic acids are increased in animals reared on Δcfa bacteria likely because these same FAs are the substrates for cyclopropane synthase and therefore accumulate in the absence of the CFA enzyme. In addition, we found that a specific subset of PUFAs was dramatically enriched in animals fed Δcfa *E. coli*, some of which have been previously observed in fatty acid desaturation mutants that accumulate palmitoleic acid (PMID: 17435249). We discuss these results in the manuscript (Lines 121-137) and have included a new figure panel in the main text (Figure 2C) and new extended data figures (Extended Data Figures 1 and 2).

Do E. coli lipids integrate into C. elegans cells in WT vs cfa mutant? A full picture of the complexity of these microbe-host lipid interactions is missing or buried in the metabolomic raw data from the experiment.

Response: As mentioned above, we profiled lipid derivatives that are absent from *E. coli* and therefore represent host metabolism, including phosphatidylcholines and *N*-acyl glycolyglycerophosphoethanolamines (PMID: 22922101 and 35145075), several of which incorporate cyclopropane fatty acids (new Extended Data Figure 2).

Minor Comments

1. *In the discussion, line 301, it is listed that the link to human biology in the diet is cyclopropane lipids are more often found in animal products. This links to a paper showing silage grain fed cattle have cyclopropane in their cheese compared to cattle allowed to forage on grass, and used*

as a biomarker for this. To me this misses a major point. One, is that you can find cyclopropane lipids widely within humans from their own microbiome and secondly they are also found in plants. The authors should make it clear that these cyclopropane lipids that promote lipid desaturation in C. elegans are not becoming more abundant in the diet because of animal products per say but the grains they are fed and they are also common in plants and human endogenous microbiome.

Response: We have expanded our citations of relevant literature as suggested by this reviewer. Our literature survey indicates that dairy and meat products derived from animals fed fermented grains are the most abundant sources of cyclopropane lipids in the Western diet (PMID: 30451036). Based on our database searches, cyclopropane lipids have not been extensively analyzed in humans. We have edited the discussion to more clearly state that animals obtain cyclopropyl lipids from ensilaged (fermented) grains, and these bacteria-derived lipids can be passed up the food chain, primarily in the form of dairy and meat products (Lines 327-331).

While it is true that many gut bacteria harbor cyclopropane synthase genes, the extent to which gut microbes synthesize cyclopropane lipids in the context of the gut has not been investigated in humans. We agree that this is a very interesting area for future studies. Intriguingly, interaction of *cis*-9,10-methylenehexadecanoic acid with a mammalian G-protein-coupled receptor was discovered by *in vitro* screening (PMID 31378678), suggesting that cyclopropyl fatty acids may act as signaling molecules in humans. We added mention of these points in the revised discussion (see Lines 318-326).

No food entry on FooDB has a confirmed measurement of either LBA or its positional isomer derived from oleic acid, dihydrosterculic acid (DHSA, *cis*-9,10-methyleneoctadecanoic acid). LBA is present in MiMeDB (MMDBc0060386), but the entry is limited. DHSA is purportedly present in a variety of foods when searching on HMDB. However, when searching in FooDB, DHSA is not present, nor do we think it is likely that it would be found in, e.g., apples or bananas. There are several cyclopropene fatty acids found in plants, e.g., in the *Sterculia* genus, but the corresponding cyclopropane fatty acids are relatively scarce.

2. In general, more should be done in the discussion to put the findings into context for mammalian biology; what are the limitations in applicability to other organisms aside from C/ elegans of these discovered interactions

Response: We thank the reviewer for this suggestion and have sharpened the discussion to reflect that gut microbes may produce cyclopropyl lipids that function as signaling molecules (Lines 318-339).

3. Figure 1D the volcano plot is poorly labeled, which metabolites there are most up and downregulated (label this)

Response: We have annotated the most prominent features in the volcano plot with molecular formulae, as requested.

4. Can you comment on the localization of Fat7-GFP to the polar ends of C. elegans after becyp#1 addition? Is this biologically significant? The expression is not homogenous as it is predicted to be in figure 1B

Response: Figure 1B is a simple schematic, in which the WT animal is depicted with the ends slightly green, whereas the *acdH-11* mutant is depicted with a more intense green extending throughout the body. These schematics roughly reflect the observed patterns of GFP expression (see Figure 2D). FAT-7 is expressed in the intestines and other tissues of *C. elegans* (see, e.g.,

PMID: 28818938), consistent with the images. The biological significance of FAT-7 expression in different tissues has not been studied in detail; however, delineating cell-type-specific functions of metabolic enzymes is an important future goal and *C. elegans* is a particularly well-suited model system for this.

5. The staining in Figure 2 in *Fat7* is different than Figure 1 how do you explain this within the model of *becyp#1* addition driving *Fat7* is it tissue specific? There is also no mention of the localization of the *Fat7* signal in Figure 3

Response: We thank the reviewer for the attention to detail. In Figure 2, the animals were grown at 20 °C. In Figures 1 and 3, the animals were grown at 25 °C but were supplemented with compounds, as indicated in the Figure legends. Expression of FAT-7::GFP in Figure 3 is consistent with Figure 1 and more generally consistent with expression at 25 °C. Control conditions were included in parallel for every independent experiment due to natural biological and experimental variation. However, FAT-7::GFP expression in WT animals is distinct from the expression in *acdh-11* animals, potentially due to cell-type specific build-up of *becyp#1* in *acdh-11* animals. We now comment on this observation in the Results section (Lines 149-152).

6. In Figure 4A which enzyme converts oleic acid to PUFAs? Are these enzymes missing because they are unknown?

Response: The FAT-2 desaturase converts oleic acid to linoleic acid, the precursor for the major PUFAs in *C. elegans*. We cited an excellent review detailing the biosynthesis of PUFAs (reference 39, Lines 66-67). We did not include this portion of the pathway in the figure because we want to focus on the generation of oleoyl-CoA by FAT-6 / FAT-7.

Reviewer #2 (Remarks to the Author):

This is a fascinating study that breaks new ground in the regulation of fatty acid desaturation. The ratio of saturated to unsaturated fats in membranes of C. elegans, which is important to maintain membrane fluidity in changing environments, has been shown to involve several nuclear hormone receptors regulating the Δ -9 fatty acid desaturase enzymes. This study used exquisite stable isotope and mass spec techniques to discover several ligands that activate the nuclear hormone receptor NHR-49 in response to nutritional cues. Interestingly, one of the ligands is a bacterial fatty acid metabolite and the other is an endogenously synthesized metabolite. During the course of this work, the authors discovered that one C. elegans acyl-coA dehydrogenase, acdh-11, has specialized activity that is required for the breakdown of cyclopropyl fatty acids, fatty acids that are particularly large dietary components of lab-grown C. elegans due to their abundance in the common C. elegans food E. coli OP50. The other groundbreaking finding is discovery of a unique nematode methyl transferase that likely evolved via horizontal gene transfer between bacteria and nematodes. This methyl transferase, FCMT-1, is required to synthesize the endogenous ligand that activates NHR-49. The experiments are thorough, carefully done and beautifully presented, and the interpretations are supported by the experimental data. I learned a lot about specialized fatty acid breakdown pathways reading this paper, it truly represents an important addition to lipid metabolism literature.

Response: We greatly appreciate the positive evaluation of our manuscript.

REVIEWERS' COMMENTS

Reviewer #1 (Remarks to the Author):

The authors of the manuscript have gone above and beyond to address my comments, and the manuscript has improved greatly. The data presented are sound and conclusions are supported by the data presented. There has been a noticeable effort to make the data more relevant to mammalian biology, which will interest the broad readership of this journal.

I have no further comments on the manuscript, all my points have been addressed.

Reviewer #2 (Remarks to the Author):

As I stated in my previous review, this is an excellent study that breaks new ground in the field of lipid metabolism. I believe the authors have addressed the concerns of the other reviewer adequately.